# Effects of alkalinity and salinity at low and high light intensity on hydrogen isotope fractionation of long-chain alkenones produced by *Emiliania huxleyi*

Gabriella M. Weiss[1], Eva Pfannerstill[1], Stefan Schouten[1,2], Jaap S. Sinninghe Damsté[1,2], Marcel T.J. van der Meer[1]

[1]Department of Marine Microbiology and Biogeochemistry, NIOZ, Royal Netherlands Institute for Sea Research and Utrecht University, Den Burg, 1790 AB, The Netherlands
[2]Department of Earth Sciences, Faculty of Geosciences, Utrecht University, Utrecht, postal code, The Netherlands

*Correspondence to*: Gabriella M. Weiss (gabriella.weiss@nioz.nl)

**Abstract.** Over the last decade, hydrogen isotopes of long-chained alkenones have been shown to be a promising proxy for reconstructing paleo sea surface salinity due to a strong hydrogen isotope fractionation response to salinity across different environmental conditions. However, to date, the decoupling of the effects of alkalinity and salinity, parameters that co-vary in the surface ocean, on hydrogen isotope fractionation of alkenones has not been assessed. Furthermore, as the alkenone-producing haptophyte, *Emiliania huxleyi*, is known to grow in large blooms under high light intensities, the effect of salinity on hydrogen isotope fractionation under these high irradiances is important to constrain before using $\delta D_{C37}$ to reconstruct paleosalinity. Batch cultures of the marine haptophyte *E. huxleyi* strain CCMP 1516 were grown to investigate the hydrogen isotope fractionation response to salinity at high light intensity and independently assess the effects of salinity and alkalinity under low light conditions. Our results suggest that alkalinity does not significantly influence hydrogen isotope fractionation of alkenones, but salinity does have a strong effect. Additionally, no significant difference was observed between the fractionation responses to salinity recorded in alkenones grown under both high and low light conditions. Comparison with previous studies suggests that the fractionation response to salinity in culture is similar under different environmental conditions, strengthening the use of hydrogen isotope fractionation as a paleosalinity proxy.

## 1 Introduction

Ocean circulation plays a strong role in global heat and moisture transport (Rahmstorf, 2002) and is controlled in part by differences in temperature and salinity, known as thermohaline circulation. Therefore, knowing these parameters is important to reconstruct ocean circulation in the geological past, which leads to a more robust understanding of our climate system. A number of valuable proxies exist to reconstruct sea surface temperature, for example, $\delta^{18}O_{foram}$ (Emiliani, 1955), Mg/Ca (Elderfield and Ganssen, 2000), $TEX_{86}$ (Schouten et al., 2002), $U_{37}^{K'}$ (Brassell et al., 1986), and LDI (Rampen et al., 2012). However, there are currently very few proxies for reconstructing sea surface salinity (SSS).

Evaporation, precipitation, continental runoff, and ice melt cause changes in seawater salinity, thereby influencing ocean circulation. The isotopic ratios of oxygen ($\delta^{18}O$) and hydrogen ($\delta D$) of water are strongly tied to these environmental parameters (Craig and Gordon, 1965). Increasing evaporation causes both enrichment in heavy isotopes (Clark and Fritz, 1997) and increase in surface water salinity. The resulting water vapor has a depleted isotopic signature (Clark and Fritz, 1997) and the longer the water stays in vapor form, the more depleted the isotopic signature of the vapor becomes as relatively enriched water precipitates first. Therefore, a depleted isotopic signature is found for most precipitation-fed rivers and lakes (i.e. meteoric waters). As these waters drain into the ocean and mix with seawater, the sea surface salinity is lowered, as is the water isotope value. This leads to a strong linear correlation between $\delta^{18}O_{water}$ values and salinity in ocean water and therefore the $\delta^{18}O_{water}$ is a suitable proxy for sea surface salinity. However, the slope of the correlation varies in space (ocean region) and time (Duplessy et al., 1993; Mashiotta et al., 1999), severely complicating reconstructions of ancient $\delta^{18}O_{water}$-S relationships and, thus, paleosalinity reconstructions. Therefore, constraining the correlation between $\delta^{18}O_{water}$ values and S currently poses a challenge in attempts to extract reliable palaeosalinity estimates from inferred $\delta^{18}O_{water}$ values.

Over the last decade, culture studies have shown that the hydrogen isotopic ratios of long chain alkenones ($\delta D_{C37}$), biomarkers of Haptophyte algae from the order Isochrysidales (Volkman et al., 1980), correlate with the hydrogen isotopic ratios of the water in which the algae grow ($\delta D_{H2O}$) (Englebrecht and Sachs, 2005; Paul, 2002), which in turn is correlated with salinity (Craig and Gordon, 1965). In addition to the observed relationship between $\delta D_{C37}$ and $\delta D_{H2O}$ values, biological hydrogen isotope fractionation has been shown to decrease with increasing salinity, thereby amplifying the salinity to seawater $\delta D$ relationship of alkenones grown in culture (Schouten et al., 2006; Wolhowe et al., 2009; M'Boule et al., 2014; Chivall et al., 2014; Sachs et al., 2016). Therefore $\delta D_{C37}$ has been proposed as an appropriate proxy for reconstructing sea surface salinity (SSS) (Englebrecht and Sachs, 2005; Schouten et al., 2006). For example, $\delta D_{C37}$ values measured on alkenones extracted from Mediterranean sapropel S5 shows similar trends to $\delta^{18}O$ measured on planktonic foraminifera and suggests a salinity decrease of 6 in the Eastern Mediterranean at the onset of sapropel formation (van der Meer et al., 2007). $\delta D_{C37}$ values from Panama Basin sediments show changes in amount of runoff from the San Juan River, aligning well with instrumental data and even tracking glacial to interglacial changes in salinity (Pahnke et al., 2007). Salinity changes in the Agulhas Current system were also recorded by changes in $\delta D_{C37}$ values during glacial termination I and II and from the last glacial maximum into the Holocene, which align with $\delta^{18}O_{foram}$ values from the same region (Simon et al., 2015; Petrick et al., 2015; Kasper et al., 2014). Leduc et al. (2013) show a divergence in estimates of sea surface salinity between $\delta D_{C37}$ derived and planktonic foraminifera derived proxies ($\delta^{18}O_{sw}$ and Ba/Ca ratios) across the Holocene in the Gulf of Guinea, which are attributed to differences in the isotopic ratios of rainfall over the time period.

Although a relationship between $\delta D_{C37}$, $\delta D_{H2O}$ and fractionation with salinity has been observed in culture and some paleostudies show promising results, this relationship is not always found in nature. Häggi et al. (2015) did not find a significant relationship between $\delta D_{C37}$ values and salinity in suspended particulate matter from the Amazon Plume. In the

Chesapeake Bay Estuary (USA), $\delta D_{C37}$ values in sediments relate to $\delta D_{C37}$ values from suspended particulate matter filters and $\delta D_{H2O}$ values, but fractionation does not show a relationship with salinity (Schwab and Sachs, 2011). Nelson and Sachs (2014) also tested the use of $\delta D_{C37}$ in North American lakes covering a salinity range of 10-133. In these North American lakes, there is a relationship between $\delta D_{H2O}$ and $\delta D_{C37}$, but no trend between fractionation and salinity (Nelson and Sachs,

2014). These environmental datasets suggest that there are other factors affecting hydrogen isotope fractionation, which complicate the use of $\delta D_{C37}$ as a salinity proxy. Indeed, culture studies have indicated that hydrogen isotope fractionation can be influenced by a number of parameters, i.e., growth rate (Schouten et al., 2006; Wolhowe et al., 2009, Sachs and Kawka, 2015), growth phase (Chivall et al., 2014), species composition (M'Boule et al., 2014; Chivall et al., 2014), and irradiance (van der Meer et al., 2015). When the hydrogen isotope ratios of both the C37:3 and C37:2 alkenones are integrated (van der

Meer et al., 2013), the effect of temperature on hydrogen isotope fractionation has been shown to be negligible on the $\delta D_{C37}$ SSS proxy, eliminating one impeding factor (Schouten et al., 2006). Growth rate and irradiance have also been proven to influence total carbon isotope fractionation of alkenones used as a $p$CO$_2$ proxy (Pagani, 2014 and references therein). Both of these factors are related and seem to play a significant role for isotopic fractionation of alkenones, and the effects remain to be completely understood.

The effect of alkalinity on hydrogen isotope ratios and fractionation has not yet been tested. The effect of alkalinity on hydrogen isotope fractionation is unknown because some of the culture experiments (Schouten et al., 2006; M'Boule et al., 2014; Chivall et al., 2014) investigating hydrogen isotopes from alkenones created media of different salinities by evaporation, which changed alkalinity together with salinity in the culture media. In the natural environment, precipitation and evaporation do not only influence salinity, but also affect the total alkalinity ($A_T$) of the surface ocean. In fact, a strong

positive linear correlation between $A_T$ and salinity is observed in surface ocean waters (Millero et al., 1998; Lee et al., 2006), and, on top of that, large coccolithophore blooms can bring about a significant decline in surface water $A_T$ (Anning et al., 1996). Alkalinity is essentially the ability of water to neutralize acid, which is linked to the amount of $H^+$. $H^+$ is readily exchanged between extracellular and intracellular water, therefore, the amount of $H^+$ could potentially effect the hydrogen isotope composition of intracellular water, which is a source of hydrogen for synthesis of organic compounds. It is, therefore,

crucial to decouple the effects of salinity and alkalinity and assess how each effect hydrogen isotope fractionation independently.

Furthermore, culture work has shown light intensity to have a strong effect on $\alpha_{C37}$ at light intensities below 200 µmol photons m$^{-2}$s$^{-1}$, but not above (van der Meer et al., 2015). However, some of the culture studies that reported a strong correlation between hydrogen isotope fractionation and salinity were performed at relatively low light intensities (Wolhowe

et al., 2009; M'Boule et al., 2014; Chivall et al., 2014). Since algal blooms occur under high light conditions in surface waters across the globe (Nanninga and Tyrrell, 1996; Holligan et al., 1993), and hydrogen isotope fractionation is less variable at high light conditions (van der Meer et al., 2015), the effect of salinity on hydrogen isotope fractionation at high light intensity needs to be studied to better understand the potential effect of salinity on alkenones synthesized in nature.

Here we addressed these two issues by using batch cultures of the haptophyte algae *Emiliania huxleyi* in experiments where alkalinity was varied independently of salinity and where salinity was varied under high light conditions.

**2 Materials and Methods**

**2.1 Media and Culture Conditions**

Two separate batch culture experiments were conducted: 1) to assess whether alkalinity affects hydrogen isotope fractionation between alkenones and growth water ('alkalinity/salinity' experiment) and 2) to examine if the fractionation-salinity relationship seen in previous culture experiments still holds under high light conditions ('high light' experiment). A no longer calcifying strain of *E. huxleyi*, CCMP 1516, was used in these batch cultures. Because the effects of alkalinity on hydrogen isotope fractionation were being assessed, a non-calcifying strain was chosen to avoid significant changes to the

alkalinity of the media caused by the organisms, changes that have previously been shown to occur during large blooms of *E. huxleyi* (Holligan et al., 1993).

Media for all experiments was made using filtered North Sea water with added nutrients, trace metals and vitamins following the method for F/2 medium (Guillard and Ryther, 1962). Medium was diluted with ultrapure water to a salinity of approximately 25 and NaCl was added to achieve higher salinities. Salinity was measured using a VWR CO310 Portable

Conductivity, Salinity and Temperature Instrument.

The alkalinity/salinity experiments consisted of batch cultures with a salinity range of 26 – 42 and constant $A_T$ of 2.44 mM and batch cultures of salinity 34 and $A_T$ between 1.44 and 4.6 mM. For batches where alkalinity was changed, pH was kept constant (7.9 ± 0.07). $NaHCO_3$ and $Na_2CO_3$ were added to bring the medium to an $A_T$ of 2.44 mM, an average value for open ocean waters, which typically fall between 2.1 and 2.5 mM in the modern day ocean (Ilyina et al., 2009; Takahashi et

al., 1981). Concentrated HCl was added to reduce alkalinity of the medium to 1.44 mM, and bubbling with air for 48 h allowed for equilibration of $CO_2$ with the atmosphere following the method of Keul et al. (2013). To increase alkalinity of the medium, $NaHCO_3$ and $Na_2CO_3$ were added to achieve $A_T$ of 3.3 and 4.6 mM, respectively. Alkalinity was determined by titration with 0.1 M HCl and calculated using Gran plots (Gran, 1952; Johansson et al., 1983; Hansson and Jagner, 1973). Temperature was a constant 15˚C and light intensity was consistently kept at 75 µmol photons $m^{-2}s^{-1}$ using cool white

fluorescent light, with a light:dark cycle of 16:8 h.

The high light experiment was performed at five different salinities, from 25 – 35, under a light intensity of 600 µmol photons $m^{-2}s^{-1}$ using cool white fluorescent light, with a light:dark cycle of 16:8 h, and constant temperature of 18.5˚C and constant alkalinity. All batch culture experiments were performed in triplicate. Cultures were transferred to new medium five times prior to starting the experiment to remove any possible memory effects from the original stock culture and adapt the

30 algae to the desired experimental conditions. An Accuri C6 flow cytometer was used to count cell concentrations daily to calculate growth rate over the length of the experiment. Growth rate was calculated as the slope of the linear fit of the natural logarithm of cell density (ln[cell density]) in the exponential part of the growth curve. Cells were harvested during

exponential growth when cell abundance reached >$10^6$ cells mL$^{-1}$ to prevent effects of shading or reduced nutrient content of the medium by the haptophytes (10-12 days). Cultures (600 mL for alkalinity/salinity experiment and 150 mL for high light experiment) were filtered over 0.7 µm GF/F filters to collect organic material and medium was subsequently collected following filtration to determine δD of the growth water.

## 2.2 Water Isotope Analysis

Hydrogen isotopic ratios of the medium ($\delta D_{H2O}$) were measured on water collected prior to the experiment and after the experiment concluded. $\delta D_{H2O}$ was measured using elemental analysis-thermal conversion-isotope ratio monitoring mass spectrometry (EA/TC/irMS) (see Schouten et al., 2006). 1 µL of sample water was injected at least 10 times during a single analytical run. $\delta D_{H2O}$ values were corrected to an in-house North Sea (5‰) standard, which was calibrated against VSMOW and VSLAP.

## 2.3 Alkenone Analysis

Following filtration, filters were freeze-dried and extracted ultrasonically five times for 10 min each time using DCM/MeOH (2:1) to obtain total lipid extracts (TLE). TLEs were then separated into three fractions over $Al_2O_3$ column using hexane/DCM 9:1 (v:v) to elute the apolar fraction, hexane/DCM 1:1 (v:v) to elute the ketone (alkenone) fraction, and DCM/MeOH 1:1 (v:v) to elute the polar fraction. Ketone fractions were run on a gas chromatograph coupled to a flame ionisation detector (GC-FID) to determine alkenone concentrations prior to running on GC/TC/irMS to measure compound specific hydrogen isotope ratios ($\delta D_{C37}$). Both the GC-FID and the GC/TC/irMS were equipped with an Agilent CP-Sil 5 column (25 m x 0.32 mm internal diameter; film thickness = 0.4 µm). GC temperature programs were the same as discussed in M'Boule et al. (2014). The $H_3^+$ factor was measured daily on the GC/TC/irMS prior to running samples; values ranged between 2.8 and 2.9 ppm mV$^{-1}$ for the alkalinity/salinity experiments and 5.4 and 5.5 ppm mV$^{-1}$ for the high light experiments. A Mix B standard (supplied by A. Schimmelmann, Indiana University) was run to assess machine accuracy on a daily basis and samples were only run when standard deviation and error of the Mix B standard were less than 5‰. Samples were measured in duplicate and squalane was co-injected with each analytical run to monitor quality of runs; average value for squalane co-injected with high light experiment samples was -164.8‰ with standard deviation of 2.2 and -163.4‰ with standard deviation of 2.7 when co-injected with the alkalinity/salinity experiment samples. All $C_{37}$ alkenone peaks were integrated as a single peak and values are thus reported as the combined values of the C37:2 and C37:3 alkenones (van der Meer et al., 2013). The isotopic fractionation of alkenones compared to media is expressed as $\alpha_{C37}$ and calculated using the equation:

Eq. (1): $\alpha_{C37} = \dfrac{\delta D_{C37} + 1000}{\delta D_{H2O} + 1000}$

**2.4 Statistics**

Analysis of covariance (ANCOVA) was applied to test if a significant difference exists between equations of the linear regression models representative of the $\alpha_{C37}$-salinity relationship between this study and previous culture studies of *E. huxleyi*. All statistical analyses were run in R using the R stats package.

**3 Results**

A no longer calcifying strain of *E. huxleyi* was grown under high light conditions over a salinity range from 25 – 35 with constant alkalinity in our high light experiment, and low light conditions over a salinity range from 26 – 42 with an alkalinity range of 1.4 – 4.6 mM in our alkalinity/salinity experiment. Changes in salinity, alkalinity and pH were relatively small over the course of the batch culture experiments (Table 1). Changes in $\delta D_{H2O}$ values of the culture media sampled prior to

10 beginning the experiments and at the end of the experiments were minimal, generally <0.4‰ (Table 1). Slightly larger changes in $\delta D_{H2O}$ values occurred for media of the alkalinity/salinity experiments, but still <1.3‰ (Table 1) and were therefore ignored. Since the salinity of the media of was not altered by evaporation but by addition of NaCl, in contrast to previous culture studies (Schouten et al., 2006; M'Boule et al., 2014; Chivall et al., 2014), $\delta D_{H2O}$ values were not correlated with salinity due to different procedures for creating of the media (Figure 1). Furthermore, the media was created separately

for each experiment, causing differences between the original $\delta D_{H2O}$ values (Figure 1). $\delta D_{C37}$ values ranged from -230.9‰ to -197‰ across all experiments, with more depleted values at lower salinities (Table 1). Growth rates (μ) ranged from 0.65 – 0.93 $d^{-1}$ for the alkalinity/salinity experiment and 0.54 – 0.67 $d^{-1}$ for the high light experiment, with lower growth rates at higher salinities (Table 1). Growth rate is weakly correlated with $\alpha_{C37}$ for both experiments (Figure 3). Higher $\alpha_{C37}$ values occur at higher salinities for both the alkalinity/salinity and high light experiments. A strong linear relationship between $\alpha_{C37}$

values and salinity was observed in both experiments (Figure 2a,b): for the high light experiment, $\alpha_{C37}=0.002S + 0.7408$ ($R^2$ 0.92, n=14; p<0.001), and for the alkalinity/salinity experiment, $\alpha_{C37}=0.0026S +0.7098$ ($R^2$ 0.85, n=24; p<0.001). For the alkalinity/salinity experiment, $\alpha_{C37}$ remains relatively constant (0.799±0.003) over the range of alkalinity, but covers a range of 0.776 – 0.824 at constant alkalinity (Table 1, Figure 2C).

**4 Discussion**

**4.1 Impact of alkalinity and light**

In the alkalinity/salinity experiment, $\alpha_{C37}$ values changed from 0.776 to 0.824 over a salinity range of 26 to 42 and constant alkalinity, and remained constant at 0.799±0.003 at alkalinities ranging from 1.4 to 4.6 (Figure 2c). This shows that alkalinity, in contrast to salinity, does not affect hydrogen isotope fractionation of non-calcifying *E. huxleyi*. We note, however, that this experiment was performed with a no longer calcifying strain of *E. huxleyi*, and results might be different

when haptophytes are calcifying since calcification may be impacted by alkalinity, which in turn could have consequences

for other intracellular processes. At constant alkalinity over a range of salinity, we see a 2.6‰ change in fractionation per salinity unit, confirming that salinity does indeed have an effect on hydrogen isotope fractionation between alkenones and growth water (Schouten et al., 2006; M'Boule et al., 2014; Chivall et al., 2014; Sachs et al., 2016).

Alkenones synthesized by haptophytes growing at different salinities under high light (600 µmol photos m$^{-2}$ s$^{-1}$) show a strong correlation between $\alpha_{C37}$ values and salinity. This unambiguously shows that there is also a strong correlation between salinity and hydrogen isotopic fractionation in alkenones at high light intensities, as encountered in the surface layers of the ocean. Ocean surface light levels span a range from zero to over 800 PAR (over 1600 µmol photons m$^{-2}$ s$^{-1}$) (Frouin and Murakami, 2007), and haptophytes not believed to be photoinhibited and primarily bloom at light intensities above 500 µmol photons m$^{-2}$ s$^{-1}$ (Nanninga and Tyrrell, 1996). Furthermore, *E. huxleyi* has been shown to adapt to different light conditions by expressing different genes under high and low light conditions (Rokitta et al., 2012), showing that growth is possible under different light regimes.

The slope of the α-salinity correlation, or fractionation response per unit salinity, is statistically similar ($p < 0.05$) for both the alkalinity/salinity and the high light experiments, based on analysis of covariance (ANCOVA) between the linear regression models fit to each dataset. There is a weak negative correlation of growth rate (µ) with fractionation for both the alkalinity/salinity and high light experiments, $\alpha = -0.0692\mu + 0.8557$ ($R^2 = 0.25$, $n = 24$, $p < 0.05$) and $\alpha = -0.1257\mu + 0.8776$ ($R^2 = 0.35$, $n = 14$, $p < 0.05$), respectively (Figure 3), which aligns with findings of Sachs and Kawka (2015) who report a negative correlation between growth rate and fractionation, albeit a stronger relationship. Growth rate is also negatively correlated with salinity in both experiments (Table 1; Figure 3), which is consistent with earlier work of Schouten et al. (2006). However, our results show a direct effect of salinity on both growth rate and fractionation, suggesting the correlation between growth rate and fractionation might be largely indirect.

## 4.2 Comparison with previous studies

We performed a statistical comparison using ANCOVA between the different $\alpha_{C37}$-salinity relationships for previous *E. huxleyi* cultivation experiments (Schouten et al., 2006; M'Boule et al., 2014; Sachs et al., 2016; Table 2) and our experiments. Sachs et al. (2016) report δD values for individual alkenones, thus we used a weighted mean average of the δD$_{C37:3}$ and δD$_{C37:2}$ values to compare with other results reporting integrated δD$_{C37}$ values. The slopes of the $\alpha_{C37}$-salinity relationships are not statistically different from each other ($p > 0.05$), with the exception of three comparisons: Sachs et al. (2016) was statistically different ($p \leq 0.05$) from Schouten et al. (2006), M'Boule et al. (2014) and the Alkalinity/Salinity experiment (Table 2). A possible explanation for the statistical difference between the $\alpha_{C37}$-salinity relationship of Sachs et al. (2016) and the other three experiments could be due to the fact that the Sachs et al. (2016) experiment was conducted using chemostats with a controlled growth rate, whereas, the other experiments were batch culture experiments where growth rate varied. Growth rate has been shown to effect hydrogen isotope fractionation of alkenones (Schouten et al., 2006; Wolhowe et al., 2009; Sachs and Kawka et al., 2015), and therefore could account for the difference between reported

fractionation responses to salinity. Although the slopes are statistically similar (p > 0.05), different strains used in the experiments, and the individual experiments themselves (i.e., conducted by different labs using different techniques) do likely play a large role in the observed differences between slopes. Furthermore, the intercepts of the regression models applied to the $\alpha_{C37}$-salinity relationships for the *E. huxleyi* culture data are all significantly different (p ≤ 0.05), i.e. the

absolute fractionation differs between the different studies, except for the relationship reported by M'Boule et al. (2014) and our high light experiment. These differences in intercept may be explained by a number of potential factors. One explanation could be due to the different strains of *E. huxleyi* used in the experiments, as each strain would respond in a similar fashion to salinity changes, but fractionate to a different extent, similar to differences seen between species (Schouten et al., 2006; Chivall et al., 2014; M'Boule et al., 2014). This could be due to differences in fractionation between intra- and extracellular

sources of hydrogen or differences in lipid synthesis rates. Another explanation for part of the discrepancies in intercepts could be analytical differences between laboratories, i.e. small offsets in measured absolute values of $C_{37}$ alkenones.

With the exception of the high light experiment, all other culture experiments with *E. huxleyi* being discussed here were grown at light intensities between 50-300 µmol photons $m^{-2}$ $s^{-1}$ (Schouten et al., 2006; M'Boule et al., 2014; Sachs et al., 2016; alkalinity/salinity experiment). The fact that the strong $\alpha_{C37}$-salinity response is also identified in *E. huxleyi* grown at

high light conditions is important for understanding the influence of light and depth effects (i.e., van der Meer et al., 2015; Wolhowe et al., 2015) on the $\alpha_{C37}$ of alkenones preserved in the sedimentary record. Van der Meer et al. (2015) suggested that at light intensities above 200 µmol photons $m^{-2}$ $s^{-1}$, $\alpha_{C37}$ responds differently to changes in light intensity than below, with a larger reported range in fractionation values at light intensities below 200 µmol photons $m^{-2}$ $s^{-1}$. Wolhowe et al. (2015) also show this trend in $a_{C37}$ measured on alkenones in suspended particulate matter from the Gulf of California and Eastern

Tropical North Pacific. Krumhardt et al. (2016) indicate that although haptophyte indicative pigments were high below surface water layers in the subtropical North Atlantic, they were also abundant in the upper 30 m of the water column, especially during spring. Based on these findings and the UK'37 core-top calibration, we can be confident that alkenones preserved in the sediments are largely reflecting surface water temperatures during the time of the year that haptophytes are known to bloom (Müller et al., 1998). Furthermore, haptophytes are thought primarily to bloom at light intensities above 500

25  µmol photons $m^{-2}$ $s^{-1}$ in the surface ocean (Nanninga and Tyrrell, 1996), leading to the conclusion that the light and depth effects discussed previously (van der Meer et al., 2015; Wolhowe et al., 2015) might not have such a large effect on the $\alpha_{C37}$-salinity response as previously believed.

### 4.3 Potential mechanisms for salinity and light responses

As mentioned above, our results show that the response in hydrogen isotopic fractionation of alkenones to salinity is

statistically similar across different *E. huxleyi* strains and under different growth conditions, including low and high light conditions. How salinity affects hydrogen isotope fractionation is still unknown, although several possible mechanisms have been proposed (e.g., Maloney et al., 2016 and references therein). The effect of salinity on hydrogen isotope fractionation seems to be a general feature recorded in alkenones, fatty acids, sterols, phytene and diploptene produced by algal

photoautotrophs (Heinzelmann et al., 2015; Schouten et al., 2006; Sachse and Sachs, 2008; Sachs and Schwab, 2011; Nelson and Sachs, 2014). Nicotinamide adenine dinucleotide phosphate (NADPH) is associated with large isotope fractionation values, larger than the fractionation between extracellular and intracellular water, but both are used as sources of H for synthesis of organic compounds (Schmidt et al., 2003). NADPH has been proposed to supply around 50% of the H eventually used for lipid production in the bacterium *Escherichia coli* (Kazuki et al., 1980), and it is presumed to be roughly the same for photosynthetic algae (Zhang et al., 2009). The cell generates NADPH either photosynthetically or via the oxidative portion of the pentose phosphate pathway (OPP pathway) (Schmidt et al., 2003; Wamelink et al., 2008). NADPH derived via ferredoxin-NADP+ reductase (FNR) in photosystem 1 (photosynthetically derived) tends to be depleted by ~600‰ in D when compared to intracellular water (Luo et al., 1991), whereas NADPH produced via the OPP pathway is also depleted compared to intracellular water, but much less than photosynthetically derived NADPH (Schmidt et al., 2003; Maloney et al., 2016). Schmidt et al. (2003) suggested that a transfer of H from NADPH generated as part of the OPP pathway causes depletion in D which is further enhanced during continued biosynthesis, meaning organic compounds containing H largely derived from metabolically reduced NADPH are characterized by depletion in D. However, this depletion is still less than what is observed for photosynthetically derived NADPH. Up or down regulation of the OPP pathway relative to other NADPH generating pathways (FNR derived, for instance) could, therefore, cause differences in the amount of D depletion of organic compounds. Up regulation of the pentose phosphate pathway observed in the bacterium *Vibrio sp.* at high salinities led to an increase in NADPH generated by the pathway for use in biosynthesis (Danevčič and Stopar, 2011). Danevčič and Stopar (2011) also found that intracellular production of L-proline, an osmoregulating amino acid, increased. The advantage of up regulating the OPP derived NADPH would be tied to this increase in L-proline, which helps continue growth and biosynthesis at higher salinities in *Vibrio* sp. A similar mechanism could be present in *E. huxleyi*, causing the metabolically reduced NADPH pool to increase relative to other pools, and possibly become a more important source of NADPH for biosynthesis if the OPP pathway exists in the same location as the site of alkenone synthesis. However, in photoautotrophic organisms, the reduction of $NADP^+$ to NADPH is also directly linked to photosystem activity (FNR) and, therefore, light intensity (Allen, 2002), and as previously mentioned, this initial reduction is characterized by a very large fractionation step (Schmidt et al., 2003; Maloney et al., 2016; Luo et al., 1991). Because of this, we would expect isotope ratios to change with different light intensities. Indeed, van der Meer et al. (2015) showed this to be the case for irradiances between 15 and 200 µmol photons $m^{-2} s^{-1}$, but the effect of changing light intensity on hydrogen isotope fractionation is much lower at light intensities >200 µmol photons $m^{-2} s^{-1}$. Furthermore, Sachs et al. (2017) showed a light effect on hydrogen isotope fractionation of C14:0 fatty acid, but no effect of light was observed on hydrogen fractionation of C16:0 and C16:1 fatty acids from the diatom *Thalassiosira pseudonana* grown over a low light range from 6 – 47 µmol $m^{-2}$ $s^{-1}$. The lack of correlation with light intensity for the longer fatty acids is explained by enzymatic reprocessing that causes further hydrogen fractionation and overwrites the light effect seen for the C14:0 (Sachs et al., 2017). This effect could also apply to alkenones, since alkenone synthesis has been linked to fatty acid biosynthesis (Volkman et al., 1980; Marlowe et al., 1984; Rontani et al., 2006). However, at high light conditions, where more photosynthetically derived NADPH is expected

to be available (e.g. our high light experiment), the same fractionation response to salinity is observed as at low light conditions (e.g. M'Boule et al., 2014; our alkalinity/salinity experiment), where less photosynthetically derived NADPH is expected. This suggests that light intensity does not directly have an effect on the predominance of photosynthetically derived versus metabolically derived, or enzymatically reprocessed NADPH used in biosynthesis, or that the up-regulation of metabolically derived NADPH with increasing salinity exerts a stronger control on hydrogen isotope fractionation than irradiance.

Another explanation for the observed significant correlation with salinity at both high and low light intensity could be that the cell synthesizes alkenones in a closed cell compartment, similar to the 'coccolith vesicle-reticular body' in which coccoliths are formed (Wilbur et al., 1963; Sviben et al., 2016), where the amount of NADPH used for biosynthesis is regulated and the fraction NADPH derived from the OPP pathway into the closed compartment increases with increasing salinity.

In addition to higher abundance of NADPH generated by the OPP pathway at higher salinities, cells could also produce more D-depleted compounds, osmolytes for instance (Dickson et al., 1982; Sachs et al., 2016 and references therein; Sachse et al., 2008; Danevčič and Stopar, 2011), which would leave the intracellular NADPH pool more enriched, and would result in D enrichment of other biosynthetic products such as alkenones. The production of DMSP, an osmolyte produced by marine microalgae, is not coupled to light intensity (van Rijssel and Gieskes, 2002), therefore, osmolyte production could be a major factor responsible for the salinity response observed over a range of light intensities. An added complication could be that cells excrete more isotopically depleted osmolytes at high salinities than at low salinities (Demidchik et al., 2014), which could leave the fraction of NADPH used for other organic compounds more isotopically enriched at high salinities. These processes are correlated with salinity, however, if NADPH plays a central role in hydrogen isotope fractionation and the reduction of $NADP^+$ to NADPH is directly coupled to photosystem activity and, therefore, light intensity, different slopes for $\alpha_{C37}$-salinity are expected for cells grown at high light and low light conditions, which is in contrast to what our results show.

Based on the compilation of *E. huxleyi* culture data, a significant relationship between hydrogen isotope fractionation and salinity is observed. However, we do not see a clear relationship between hydrogen isotope fractionation and light intensity. Due to balancing between ATP and NADPH production and consumption within the cell (Walker et al., 2014), NADPH formation dominates at high light levels, whereas ATP synthesis dominates at lower light levels (Beardall et al., 2003), leading to the idea of a larger pool of photosynthetically derived NADPH inside the cell under high light conditions, would, in turn, cause differences in hydrogen isotope fractionation during the synthesis of alkenones, but this is not what the data shows. Transhydrogenase enzyme activity can remove NADPH when in excess by reducing NAD+ to NADH using NADPH (Kim and Gadd, 2008; Zhang et al., 2009), which is associated with a large isotope fractionation effect of between 800-3500‰ (Zhang et al., 2009), leaving a relatively D enriched pool of NADPH behind. At high light conditions, an excess of NADPH is expected, and therefore, increased transhydrogenase activity, something also seen with increasing salinity (Danevčič and Stopar, 2011). However, if transhydrogenase enzyme activity is responsible for reducing the excess NADPH,

we might expect to see a difference in isotope values and a different fractionation response to salinity at high and low light conditions, which is not the case. There is, of course, the possibility that the cell could use excess NADPH for other intracellular processes or synthesis of compounds that are not being investigated or measured in our experiments, which could be light intensity dependent as well, similar to what was reported for different fatty acids by Sachs et al. (2017). This would explain why having an abundance of NADPH at high light intensities does not seem to affect hydrogen isotope fractionation of alkenones since this abundance of photosynthetically derived NADPH is being used for other processes or is counteracted by enzymatic activity (Sachs et al., 2017). A better understanding of the hydrogen isotopic composition of different relevant hydrogen pools and how the alkenone synthesis process works is required for more accurate determinations of the mechanism responsible for the strong salinity response to hydrogen isotope fractionation of alkenones.

## 5 Conclusions

Our results show that salinity has a strong effect on hydrogen isotope fractionation of alkenones in cultivated *E. huxleyi* at high light intensities. In contrast, alkalinity, although co-varying with salinity in environmental waters, does not affect hydrogen isotope fractionation between alkenones and growth water. Interestingly, we see a similar effect of salinity on hydrogen isotope fractionation at both high and low light conditions. Our present knowledge of biosynthetic mechanisms does not allow us to explain the similarity of the fractionation response to salinity at high and low irradiance in absolute terms. However, further investigation of intracellular sources and partitioning of intracellular hydrogen could allow us to explain this mechanism more accurately. The fact that light intensity is a function of depth in the water column, and the abundance of alkenones decreases with depth, the effect of lower light intensity on hydrogen isotope fractionation observed in previous studies is likely minor for sedimentary alkenones. Our results show the consistency of the hydrogen isotope fractionation response to salinity for multiple *E. huxleyi* strains grown under different conditions, and further supports the use of $\alpha_{C37}$ for reconstructing paleo sea surface salinity.

**Author Contribution**

Gabriella Weiss, Marcel T.J. van der Meer and Eva Pfannerstill designed the experiments and Gabriella Weiss and Eva Pfannerstill carried them out. Gabriella Weiss prepared the manuscript with contributions for all co-authors.

**Competing Interests**

Marcel T.J. van der Meer is an Associate Editor of Biogeosciences.

**Acknowledgements**

This study received funding from the Netherlands Earth System Science Center (NESSC) though a gravitation grant (024.002.001) from the Dutch Ministry for Education, Culture and Science. The authors would like to thank Associate Editor

Markus Kienast as well as Alex Sessions and three other anonymous reviewers for their constructive comments which helped to improve this manuscript.

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

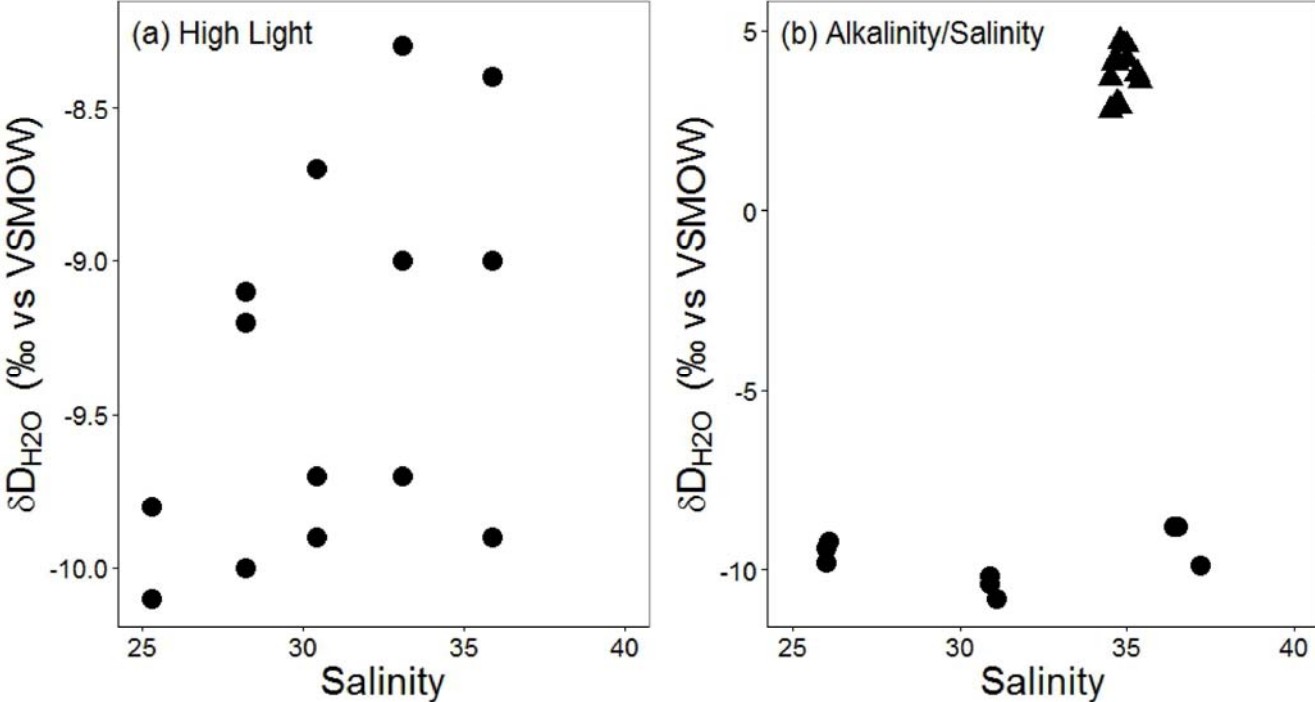

Figure 1 shows the lack of correlation between $\delta D_{H2O}$ and salinity of the culture media for both the high light (a) and the alkalinity/salinity (b) experiments. The triangles in the alkalinity/salinity plot are the $\delta D_{H2O}$ values of the media covering a range in alkalinity, whereas the circles represent the $\delta D_{H2O}$ values for the constant alkalinity media.

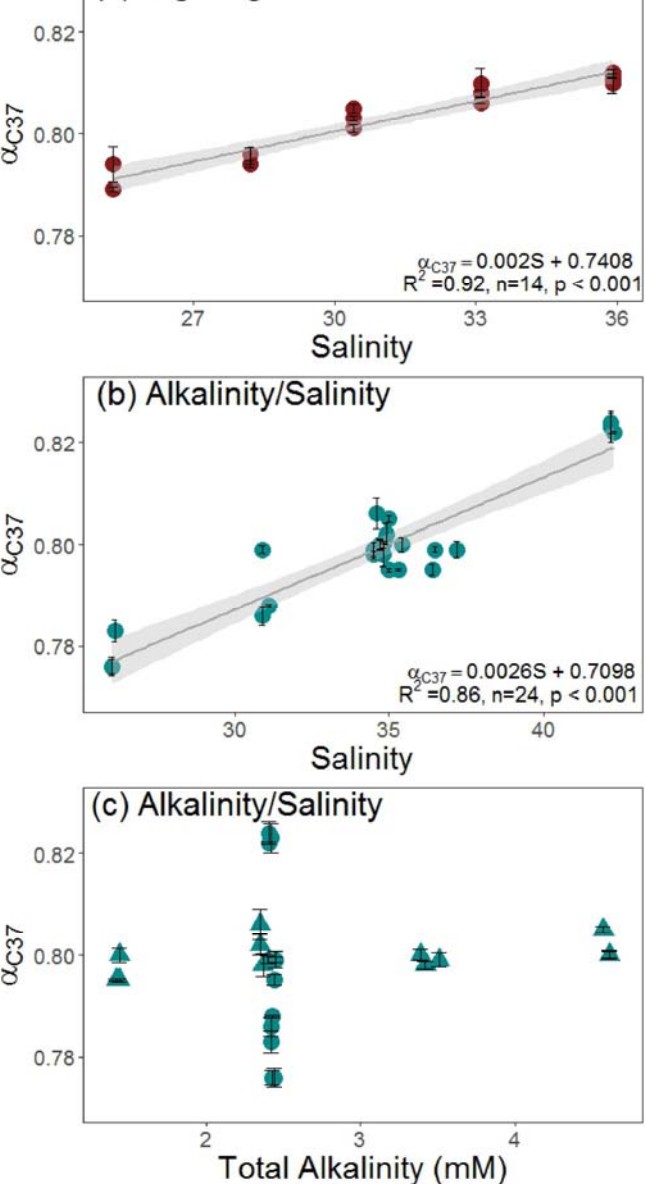

Figure 2: Hydrogen isotope fractionation factor $\alpha_{C37}$ plotted against salinity for the (a) high light experiment and (b) alkalinity/salinity experiment. The grey shading represents the 95% confidence interval for the linear regression model fit to the dataset. 2c shows hydrogen isotope fractionation factor $\alpha_{C37}$ plotted against total alkalinity for the alkalinity/salinity experiment. The triangles represent the set of batch cultures grown at constant salinity (35) over a range of alkalinity (1.4 - 4.6), and the circles represent the batch cultures grown at constant alkalinity (2.4) over a range of salinity (26-42).

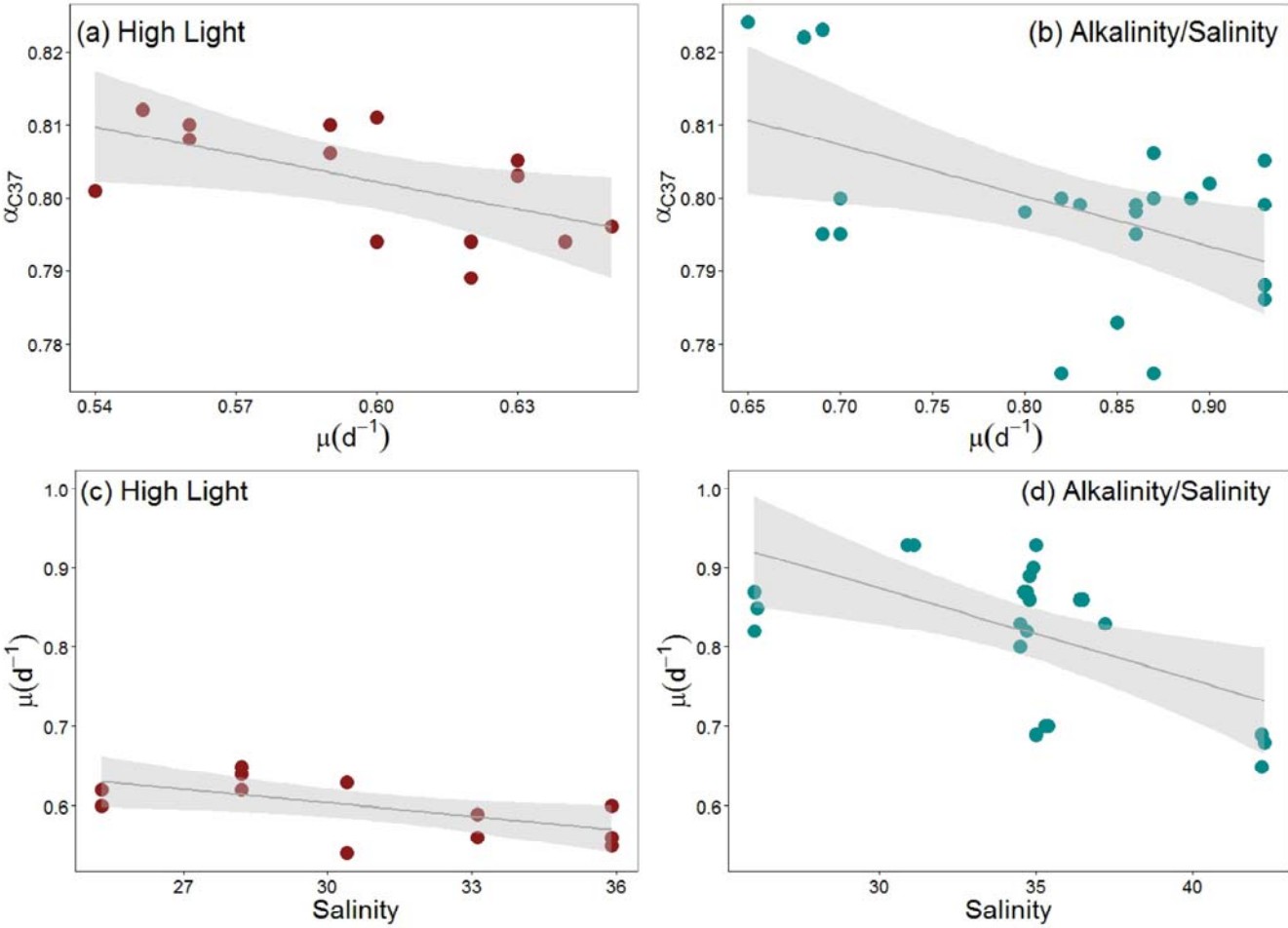

Figure 3: Relationships of growth rate ($\mu$ d$^{-1}$) with hydrogen isotope fractionation factor $\alpha_{C37}$ and salinity. The grey shading represents the 95% confidence interval for the linear regression model applied to the data. (a) Growth rate and $\alpha_{C37}$ relationship for the high light experiment shows a weak negative correlation ($R^2 = 0.35$, n = 14, p < 0.05). (b) Growth rate and $\alpha_{C37}$ relationship for the alkalinity/salinity experiment also shows a weak negative correlation ($R^2 = 0.25$, n = 24, p < 0.05). (c) Salinity and growth rate for the high light experiment ($R^2 = 0.29$, n = 14, p < 0.05). (d) Salinity and growth rate for the alkalinity/salinity experiment ($R^2 = 0.29$, n = 24, p < 0.05).

Table 1: Growth parameters and hydrogen isotope ratios of alkenones from Alkalinity/Salinity and High Light batch cultures of *Emiliania huxleyi* CCMP 1516.

| Salinity | Temperature (°C) | Irradiance (μmol photons $m^{-2}s^{-1}$) | Growth rate ($d^{-1}$) | $\delta D_{H2O}$ (‰ vs. VSMOW) initial | St. dev. | $\delta D_{H2O}$ (‰ vs. VSMOW) end | St. dev. | $\delta D_{C37}$ (‰ vs. VSMOW) | St. dev. | $\alpha_{C37}$ | Error | $A_T$ mM initial | $A_T$ mM end | pH initial | pH end |
|---|---|---|---|---|---|---|---|---|---|---|---|---|---|---|---|
| *Alkalinity and Salinity Experiment Strain CCMP1516* | | | | | | | | | | | | | | | |
| 26.0 | 15 | 75 | 0.82 | -10.0 | 1.1 | -8.8 | 1.8 | -230.9 | 1.8 | 0.776 | 0.002 | 2.39 | 2.49 | 8 | 8.8 |
| 26.0 | 15 | 75 | 0.87 | -10.0 | 1.1 | -9.6 | 1.6 | -231.1 | 1.4 | 0.776 | 0.001 | 2.39 | 2.46 | 8 | 8.7 |
| 26.1 | 15 | 75 | 0.85 | -10.0 | 1.1 | -8.5 | 2.0 | -224.3 | 2.1 | 0.783 | 0.002 | 2.39 | 2.46 | 8 | 8.7 |
| 30.9 | 15 | 75 | 0.93 | -10.8 | 1.7 | -10.1 | 2.6 | -209.1 | 0.7 | 0.799 | 0.001 | 2.38 | 2.42 | 7.9 | 7.8 |
| 31.1 | 15 | 75 | 0.93 | -10.8 | 1.7 | -10.7 | 0.9 | -220.3 | 0.2 | 0.788 | 0.000 | 2.38 | 2.47 | 7.9 | 8.7 |
| 30.9 | 15 | 75 | 0.93 | -10.8 | 1.7 | -9.6 | 1.1 | -221.6 | 1.8 | 0.786 | 0.002 | 2.38 | 2.46 | 7.9 | 8.6 |
| 36.5 | 15 | 75 | 0.86 | -8.7 | 1.1 | -8.9 | 1.8 | -207.8 | 0.5 | 0.799 | 0.001 | 2.42 | 2.48 | 7.8 | 8.7 |
| 37.2 | 15 | 75 | 0.83 | -8.7 | 1.1 | -11.1 | 1.5 | -209.3 | 1.6 | 0.799 | 0.002 | 2.42 | 2.49 | 7.8 | 8.8 |
| 36.4 | 15 | 75 | 0.86 | -8.7 | 1.1 | -8.9 | 1.4 | -212.5 | 1.1 | 0.795 | 0.001 | 2.42 | 2.45 | 7.8 | 8.6 |
| 42.2 | 15 | 75 | 0.69 | -7.4 | 1.6 | -7.4 | 1.6 | -183.1 | 2.9 | 0.823 | 0.003 | 2.38 | 2.46 | 7.9 | 8.6 |
| 42.2 | 15 | 75 | 0.65 | -7.4 | 1.6 | -8.6 | 0.9 | -182.8 | 2.3 | 0.824 | 0.002 | 2.38 | 2.45 | 7.9 | 8.6 |
| 42.3 | 15 | 75 | 0.68 | -7.4 | 1.6 | -7.3 | 1.0 | -184.4 | 0.2 | 0.822 | 0.000 | 2.38 | 2.45 | 7.9 | 8.7 |
| 35.4 | 15 | 75 | 0.7 | 3.6 | 1.2 | 3.5 | 1.6 | -197.0 | 1.4 | 0.800 | 0.001 | 1.39 | 1.5 | 7.8 | 8 |
| 35.3 | 15 | 75 | 0.7 | 3.6 | 1.2 | 4.0 | 1.7 | -201.8 | 0.2 | 0.795 | 0.000 | 1.39 | 1.45 | 7.8 | 8.5 |
| 35.0 | 15 | 75 | 0.69 | 3.6 | 1.2 | 5.6 | 2.1 | -201.5 | 0.3 | 0.795 | 0.000 | 1.39 | 1.48 | 7.8 | 8.5 |
| 34.8 | 15 | 75 | 0.86 | 4.4 | 1.2 | 5.0 | 1.1 | -198.7 | 2.2 | 0.798 | 0.002 | 2.32 | 2.42 | 7.9 | 8.6 |
| 34.6 | 15 | 75 | 0.87 | 4.4 | 1.2 | 3.8 | 1.0 | -190.2 | 3.0 | 0.806 | 0.003 | 2.32 | 2.39 | 7.9 | 8.5 |
| 34.9 | 15 | 75 | 0.9 | 4.4 | 1.2 | 4.8 | 1.7 | -194.5 | 2.1 | 0.802 | 0.002 | 2.32 | 2.39 | 7.9 | 8.6 |
| 34.7 | 15 | 75 | 0.82 | 2.3 | 1.1 | 3.8 | 1.2 | -197.1 | 1.0 | 0.800 | 0.001 | 3.32 | 3.45 | 8 | 8.6 |
| 34.5 | 15 | 75 | 0.8 | 2.3 | 1.1 | 3.3 | 1.0 | -199.8 | 0.7 | 0.798 | 0.001 | 3.32 | 3.52 | 8 | 8.8 |
| 34.5 | 15 | 75 | 0.83 | 2.3 | 1.1 | 5.1 | 1.1 | -198.1 | 1.4 | 0.799 | 0.001 | 3.32 | 3.69 | 8 | 8.7 |
| 35.0 | 15 | 75 | 0.93 | 4.1 | 1.5 | 4.2 | 1.3 | -191.6 | 0.7 | 0.805 | 0.001 | 4.58 | 4.56 | 7.9 | 8.5 |
| 34.8 | 15 | 75 | 0.89 | 4.1 | 1.5 | 1.7 | 1.5 | -197.5 | 0.6 | 0.800 | 0.001 | 4.58 | 4.63 | 7.9 | 8.6 |
| 34.7 | 15 | 75 | 0.87 | 4.1 | 1.5 | 4.0 | 1.4 | -197.0 | 0.8 | 0.800 | 0.001 | 4.58 | 4.63 | 7.9 | 8.5 |
| *High Light Experiment Strain CCMP1516* | | | | | | | | | | | | | | | |
| 25.3 | 18.5 | 600 | 0.60 | -9.9 | 1.0 | -10.1 | 0.9 | -214.3 | 3.4 | 0.794 | 0.003 | | | | |
| 25.3 | 18.5 | 600 | 0.62 | -9.9 | 1.0 | -9.8 | 1.0 | -218.3 | 0.4 | 0.789 | 0.000 | | | | |
| 28.2 | 18.5 | 600 | 0.65 | -9.3 | 1.2 | -10.0 | 0.8 | -212.1 | 1.3 | 0.796 | 0.000 | | | | |
| 28.2 | 18.5 | 600 | 0.62 | -9.3 | 1.2 | -9.2 | 1.0 | -213.1 | 0.0 | 0.794 | 0.001 | | | | |
| 28.2 | 18.5 | 600 | 0.64 | -9.3 | 1.2 | -9.1 | 1.1 | -213.2 | 0.6 | 0.794 | 0.001 | | | | |
| 30.4 | 18.5 | 600 | 0.63 | -10.7 | 1.2 | -9.9 | 0.7 | -204.9 | 0.5 | 0.803 | 0.001 | | | | |
| 30.4 | 18.5 | 600 | 0.54 | -10.7 | 1.2 | -8.7 | 1.2 | -205.6 | 0.8 | 0.801 | 0.001 | | | | |
| 30.4 | 18.5 | 600 | 0.63 | -1.7 | 1.2 | -9.7 | 1.0 | -203.1 | 0.8 | 0.805 | 0.001 | | | | |
| 33.1 | 18.5 | 600 | 0.56 | -9.0 | 1.5 | -9.7 | 1.3 | -199.6 | 0.7 | 0.808 | 0.003 | | | | |
| 33.1 | 18.5 | 600 | 0.59 | -9.0 | 1.5 | -8.3 | 0.9 | -196.8 | 2.9 | 0.810 | 0.000 | | | | |
| 33.1 | 18.5 | 600 | 0.59 | -9.0 | 1.5 | -9.0 | 1.5 | -201.1 | 0.0 | 0.806 | 0.000 | | | | |
| 35.9 | 18.5 | 600 | 0.60 | -9.5 | 0.9 | -9.9 | 0.7 | -197.2 | 0.0 | 0.811 | 0.001 | | | | |
| 35.9 | 18.5 | 600 | 0.55 | -9.5 | 0.9 | -9.0 | 1.3 | -195.1 | 0.7 | 0.812 | 0.002 | | | | |
| 35.9 | 18.5 | 600 | 0.56 | -9.5 | 0.9 | -8.4 | 1.3 | -197.2 | 1.9 | 0.810 | 0.000 | | | | |

Table 2: Linear regression equations for hydrogen isotope fractionation - salinity ($\alpha_{C37}$-salinity) relationship for a compilation of culture experiments growing different strains of *Emiliania huxleyi.*

| Reference | Strain | $\alpha_{C37}$-Salinity relationship | $R^2$ | Number of points |
|---|---|---|---|---|
| Schouten et al., 2006 | *E. huxleyi* PML B92/11 | $\alpha_{C37} = 0.0033S + 0.6928$ | 0.74 | 11 |
| M'Boule et al., 2014 | *E. huxleyi* CCMP 1516 | $\alpha_{C37} = 0.0021S + 0.7401$ | 0.80 | 20 |
| Sachs et al., 2016 | *E. huxleyi* CCMP 374 | $\alpha_{C37} = 0.0015S + 0.7770$ | 0.88 | 9 |
| Alkalinity and Salinity | *E. huxleyi* CCMP 1516 | $\alpha_{C37} = 0.0026S + 0.7098$ | 0.86 | 24 |
| High Light | *E. huxleyi* CCMP 1516 | $\alpha_{C37} = 0.0020S + 0.7408$ | 0.92 | 14 |