# Peer review of "Effects of alkalinity and salinity at low and high light intensity on hydrogen isotope fractionation of long-chain alkenones produced by *Emiliania huxleyi"

_Biogeosciences, 2017_

## Referee Comment (RC1) · Anonymous Referee #1 · 21 Sep 2017

Weiss et al. (2017) conducted two experiments using batch cultures of the haptophyte Emiliania huxleyii (strain CCMP1516) to determine 1) how alkalinity (separate from salinity) might impact hydrogen isotope ratios in alkenones and 2) if high light conditions influence the previously reported salinity-fractionation relationship in alkenones. The results indicate that the alkenone hydrogen isotope salinity-fractionation relationship is robust (and similar to previously reported relationships) regardless of alkalinity and light level, which are useful and interesting findings. I recommend this manuscript for publication but request that the authors address the following issues: 1) more information about natural light levels, 2) separate alkalinity as the sole variable as part of the analysis, 3) improve the discussion about mechanisms (if possible include additional lipid isotope and lipid concentration data to support the lack of light effect), plus a handful of technical issues listed below.

High light growth conditions in the world's oceans: The idea that most alkenones are produced in the high light of surface waters is mentioned in several places (abstract line 5; intro p.3 line19; discussion section 4.2 line 30; conclusion line 26), but without references to support this claim. As satellite imagery clearly illustrates, major coccolithophore blooms occur in highly productive (mainly coastal) areas that are seasonally growth-limited and along the equator, and presumably bloom alkenones are produced in high light conditions near the surface (but surely also deeper in the water column where light is limited due to self-shading from bloom turbidity?). As is stands, the manuscript just states that high light alkenone production likely predominates. But to strengthen your argument and support your experimental design and data interpretation, the predominance of high light alkenone production should be illustrated with references that indicate 1) alkenones are mainly produced in high light at the surface, 2) how ocean surface water light level ranges compare to light ranges in your study, 3) how surface bloom productivity compares to non-bloom conditions/areas, and 4) that these "high light" bloom alkenones are indeed exported to the sediments (more so than "low light" alkenones). Because in vast areas of the ocean, it seems the primary source of alkenones comes from fairly deep in the water column where light levels are a fraction of surface values. For instance, this has been demonstrated at BATS see Fig 1 in Krumhardt et al. 2016 (doi:10.5194/bg-13-1163-2016), at ALOHA Table 3 Prahl et al. 2005 (doi:10.1016/j.dsr.2004.12.001) and further afield Ohkouchi et al. 1999 (DOI:Âă10.1029/1998GB900024). The vast areas of the ocean with potential deep-water/lowlight alkenone production should be addressed/acknowledged with clarification of how alkenones produced in non-saturating light conditions do or do not affect the alkenone dD paleosalinity proxy in some regions. The same references used in van der Meer (2015) (in quoted text below) are helpful, and should again be used here

– and if possible, supplemented with additional references: "E. huxleyi, for instance, is thought to thrive under high light conditions, at mixed layer depths generally <30 meter (Tyrrell and Merico, 2004; Harris et al., 2005). They outcompete other algal species that suffer from photoinhibition under these conditions, a process that is apparently absent in E. huxleyi (Nanninga and Tyrrell, 1996)."

Alkalinity: Fig 2c clearly shows that there is not a relationship between total alkalinity and fractionation (alpha). However, the discussion of the alkalinity results in the low light treatments is not entirely satisfying (p.6 lines 3-8), and it can be confusing when plots include data with many changing variables that are known to influence the isotopic composition of lipids. It would be useful (even if just in a supplement) to break up the different environmental parameters. Looking only at the low light cultures grown at salinity ∼35 (12 cultures, salinity range 34.5-35.4, alkalinity range 1.39-4.58, growth rate range 0.69-0.93, alpha range 0.795-0.806) (ie, excluding the cultures that have salinities of ∼26, ∼31, ∼37, or ∼42, nearly constant alkalinity ∼2.4, but growth rate range of 0.65-0.93, and alpha range of 0.776-0.824) . . . It is noteworthy that for these 12 cultures there is a correlation between total alkalinity and specific growth rate (= 0.05(0.01) * AT + 0.676(0.04), R2 = 0.51, p = 0.0056) in addition to a correlation between specific growth rate and fractionation (= 0.03(0.01) * + 0.776(0.01), R2 = 0.47, p = 0.0081) (figure attached as example). Although this is a very small range in growth rate (∼.7 to .9) compare to a previous study of the effect of growth rate on alkenones using a combination of chemostats and continuous cultures (Sachs and Kawka, 2015), it is striking (and probably noteworthy) a) how well they are correlated and b) that it is in the opposite sense of previous 2015 study. In the end, there is still not a significant relationship between alpha and total alkalinity – could it be that the minor growth rate effect here is overwriting any alkalinity effect? It is probable that you already considered all of this and decided it didn't fit in the paper – but it would be useful for the interested reader to be able to reference in at least the supplement.

Mechanisms: It is welcome that some effort is put forth to explain the cellular mechanisms of the salinity response at both high and low light. However, this part of the discussion seems confused and needs some work by improving the organization, offering introductory or concluding summaries to help identify your main points, supporting hypotheses with literature, perhaps a schematic outlining your favorite mechanism (which one is most likely here and why?), and the issues below. A little bit of reorganization could help since it seems like the discussion tries to deal with the high light and low light salinity response in both section 4.3 and 4.4. One way to improve this is if section 4.3 can just concentrated on salinity (regardless of light), then address why a different response at high light was expected, and then finally address why there was not a strong light effect here (but more on this below). The first paragraph of section 4.3 is very long, and the main point of it is lost in the length. Maybe at the start let the reader know how many mechanisms you will cover, then at the end summarize which one seems most likely. Can you rule any out with the results from your experiment? A lot of attention is given to metabolically reduced NADPH, such as that generated through OPP. However, there are a few things to consider. NADPH can't cross organelle membranes – so p.8 lines 17-21 doesn't make sense, unless the complete OPP pathway (and final steps to the alkenone synthesis pathway?) were present in this hypothetical closed compartment. Alternatively you might invoke the import of alkenone precursors (fatty acids? Pyruvate?) used to build alkenones that reflect proportional changes in photosynthetically vs metabolically reduced NADPH pools (and would have to introduce them around p.7 line18).

Lack of light effect: p.9 line3 states "we do not see a clear relationship between hydrogen isotope fractionation and light intensity." But there is no mention of this lack of relationship in the results, where should the reader go to visualize this lack of relationship? Do you mean between the 24 low light cultures compare to the 14 high light cultures in this study? There aren't really cultures in the high light with the same growth rate and salinity as the cultures in the low light (although 3 HL cultures at S=35.9 and 12 LL cultures at S=34.5-35.4 actually do show a (very small) significant increase in alpha at the higher light level – however, that could in theory be due to the slightly saltier HL media

or the slightly lower HL growth rate rather than the higher light). More importantly, I am not convinced that there is no light effect with only 2 light levels (75 and 600), especially considering the previously observed non-linear relationship between high light in strain RCC1238 and the fact that different strains have shown different responses (van der Meer et al. 2015). Is it possible that both the 75 and 600 light levels are above the range where a response would be detected for this strain? After all, growth rate is lower in the higher light cultures. Even though (it seems) the goal of the paper is not to characterize the light effect (but to test if the salinity effect holds at high light), this part of the discussion should consider the option that if batch cultures were grown at additional light levels, then a light effect in this strain might be apparent. p.9 line 4 – "At higher light intensities, we expect a larger pool of photosynthetically derived NADPH inside the cell" – yes of course, but another thing to consider is this doesn't necessarily mean that this NADPH is available for alkenone synthesis. At high light levels cells might be working hard to dump this extra reductive power (into alkenones? or other molecules types – is there literature on algae at different light levels you can turn to?), along with reducing light harvesting capacity, which might be a reason for the lack of light effect (if there is truly a lack of light effect). It seems like cells have many different options for dealing with high light situations. Were alkenone (and other lipid) concentrations measured? Were there any large concentration differences at different light levels indicating strategies for dumping excess NADPH? p.9 line 9 – why do you think that transhydrogenase activity is increased at high light? Can you provide a reference to demonstrate that this is a response to high light in haptophytes or even just eukaryotic (or even prokaryotic) algae? Rokitta et al. (2012) doi:10.1371/journal.pone.0052212 might be helpful here. Regarding the last paragraph of section 4.4 – is there any chance hydrogen isotopes were measured on other lipids from these cultures (fatty acids, brassicasterol, phytol. . .except probably not phytol if you didn't saponify)? This information could be extremely helpful for illuminating cellular changes in response to environmental variables, and the interplay between different pools of cellular hydrogen, as it was for Sachs et al. (2017) that found different responses to different fatty acids,

phytol, and a sterol, in a diatom grown under constant salinity, temperature, and growth rate at a range of light levels. In other words, just because there wasn't a light effect in the combined C37:2 and C37:3 values doesn't mean that the unique alkenones are not reacting to light differently, or that the hydrogen isotope ratios of other cellular lipids (and lipid precursors) are not reacting to light. In this sense, measuring hydrogen isotopes in all possible lipids is a powerful tool for helping to understand these cellular mechanisms and has tremendous value beyond validating paleoproxies. In the end – I am not sure you need to devote an entire section of the discussion to explaining the lack of light effect with only two light levels (and only one lipid measurement) presented. It would be great if this section of the paper could be extended with concentration data or additional lipid isotopes if that is available.

Technical comments: Abstract: First sentence should read "Over the last decade, hydrogen isotope ratios of long chained. . .." Line 18. It would be helpful to add "at low light" (or something along those lines) to the end of the sentence ". . .and independently assess the effects of salinity and alkalinity".

Intro: p.1 line 28: add the word "and" between UK37 and LDI p.2 line 30: SPM has not yet been defined. p.3 line 2: growth phase should include also the Wolhowe et al. (2009) reference. p.3 line 7: "created media by evaporation" . . . some culture studies did not use this method: note that salt was added to ultrapure water in Sachs et al. 2017. p. 3 line 13. How would this work? Please explain why the capacity (of environmental water) to take up H+ would, in theory, impact the hydrogen isotope composition of internal cell water and fractionation during synthesis.

Methods: How was total alkalinity determined? How was salinity determined? What was the equation to calculate growth rate based on how many cell count readings over how many days? What was the size of the growth vessel for batch cultures and how many liters of culture were maintained inside? Were growth chambers swirled to prevent sedimentation? How many grams of Al2O3 were used and how many ml of solvents for purification? What were the FID and IRMS instrument settings (column, oven

temp, reactor temp, gas flow, etc)? Were external standards used to correct/calibrate reported Isodat values referenced to H2 for each run? (you could cite a previous paper with this info if the settings were the same). p.5 line 10: “difference” (not “different”)

Results: Several findings that are brought up in the discussion are not mentioned in the results (but should be) including: constant alpha over a range of alkalinities, the weak negative correlation of growth rate and fractionation for both experiments, the lack of light effect. Why not report the uncertainty on the slope and intercept of the least squares linear regressions? If you are using R for your stats, then that is easy to get using R’s “summary(lm(x∼y))”. Although only very slightly biased in this case, it is probably more responsible to report R’s Adjusted R-squared as opposed to the Multiple R-squared. In the first 2 lines of the results the ranges are reported with different dashes (“25 – 35” and then “26-42”) p.5 line 20 – “in contrast to previous culture studies” . . . this isn’t true – salt was added to fresh water in Sachs et al. 2017.

Discussion: p.6 line 19 – incorrect reference – Sachs and Kawka 2015 did not find a correlation between growth rate and salinity since growth rate was held constant using continuous cultures – this reference would be more appropriate in the proceeding sentence since they did find a correlation between growth rate and fractionation. p.7 line 6. – missing the word “to” – after the words “could be due” p. 7 lines 15-17 this doesn’t make much sense – please rephrase – are you trying to say that cell water is the same as extracellular water isotopically and NADPH is more important? p.7 line 22 – missing important reference (Luo, Y.-H., Sternberg, L., Suda, S., Kumazawa, S., Mitsui, A., 1991. Extremely low D/H ratios of photoproduced hydrogen by cyanobacteria. Plant and Cell Physiology 32, 897–900) after “Photosynthetic production causes NADPH to be depleted by ∼600‰ in D when compared to intracellular water” p.7 line 24 – missing the letter “s” in “OPP pathway cause_” p.7 line 32 – after the statement “A similar mechanism could be present in E. huxleyi, causing the metabolically reduced NADPH pool to increase relative to other pools, and possibly become a more important source of NADPH for biosynthesis” you might qualify that with something like “if the

OPP pathway is present in the same compartment as alkenone production" (since NADPH doesn't cross organelle membranes). p.8 line 2 – the better reference here is Luo et al. 1991 (provided above)

Figures: Please specify what the gray polygons are (95% confidence intervals?). Why no error bars displaying analytical uncertainty on alpha or dD? Fig.2c is useful for showing the (lack of) relationship between alkalinity and alpha – in this fashion, perhaps an additional figure that displays the (lack of) relationship between light and alpha could also help. In Fig.2c, does it make sense to include the regression line and confidence intervals here? – might be more useful to use a dashed line or some other visual aid to highlight how 2c is different than 2a and 2b rather than (or in addition to) leaving out the regression statistics. Fig. 1 might not be necessary, (useful to show that culture water did not have a relationship b/t salinity and dDwater but that is a big figure for a very minor point). Why are a handful of cultures around S=35 15‰ D-enriched relative to the other cultures in the Fig.1a?

References: p.10 line 12: Is there an extra "."? Check this and other references for common mistakes (superscripts missing in some locations, middle initials incorrect in some locations, missing italics for species names in many locations, too many words in title are capitalized in at least one reference).

[Figure]

All Salinities
All Salinities
All Salinities
Constant Salinity (35)
Constant Salinity (35)
Constant Salinity (35)
specific growth rate
alpha
Total Alk
specific growth rate
$R^2 = 0.17$
$p = 0.026$
$R^2 = 0.22$
$p = 0.012$
$R^2 = -0.035$
$p = 0.64$
$R^2 = 0.508$
$p = 0.0056$
$R^2 = 0.472$
$p = 0.0081$
$R^2 = 0.0859$
$p = 0.18$

**Fig. 1.**

---

## Referee Comment (RC2) · A. Sessions (Referee) · 22 Sep 2017

This study focuses on the relationship between salinity, alkalinity, and light intensity with H isotope fractionation. The result is to show again that salinity is an important factor, but that alkalinity and light intensity are not. I would characterize this as an important, incremental advance in our understanding of this proxy. The result is not Earth-shaking, but it is an important step forward.

The results seem quite clear and unambiguous, and the data analysis and interpreta-

tion is convincing. This is a nice study, with little to complain about. My only general comment is to question why the authors chose to describe culture conditions in terms of alkalinity rather than pH. With [DIC] fixed by equilibrium with atmospheric PCO2, alkalinity and pH are in a sense interchangeable (fixing one uniquely determines the other). Thus the same experiments could be described in terms of either parameter. Alkalinity is probably more popular among oceanographers, but pH is much more widely used among biologists. And I might argue that there is some reason to think that cellular H isotope fractionation depends more on the concentration of H+ (i.e., pH) than on the ability to consume H+ (i.e., alkalinity). So my suggestion is to at least consider describing the first series of experiments as a pH series, rather than an alkalinity series. Or maybe there is a way to gracefully do both.

I was curious why a non-calcifying strain of E. hux was chosen. Perhaps it simplifies controlling alkalinity? In any case, it would be worth a few sentences of explanation about why you chose this strain, and how it might relate to strains that are prevalent in the oceans. Is it likely to be representative of strains that produce alkenones in most marine sediments?

Section 2.1. Please tell us how you measured (or calculated) alkalinity?

Page 6, line 25. You say that you performed a statistical comparison, and then that "This showed a strong similarity between slopes..". What does strong similarity mean in statistical terms? They are indistinguishable? Given that the slopes differ between experiments by more than a factor of 2, this is probably more a statement about variability between experiments rather than a constant slope. Seems like the discussion of this 'similarity' could be a bit more nuanced. Differences of a factor of $\sim$2 would still make a huge difference in reconstructing seawater salinity, even if they are statistically indistinguishable.

Page 7, line 6. The differences in intercepts amount to a range of nearly 78‰ That does not seem (to me, at least) plausible to explain solely by interlaboratory differences.

[Figure]

Maybe modify the text to say that "part of" the differences could possibly be attributed to this.

Page 7, lines 20-35. The term "photosynthetically-derived NADPH" struck me as a little odd, especially in contrast to the more precise "pentose-phosphate pathway". Photosynthesis both produces (in photosystem I of the light reactions) and consumes (in $CO_2$ fixation of the Calvin cycle reactions) NADPH. It would thus be more precise to refer to NADPH from the "light reactions of photosynthesis", or to "ferredoxin-NADP+ reductase (FNR) in photosystem 1", etc.

Page 7, line 30-33. I like this explanation, a lot. It is the best one I have heard yet.

Page 9, line 4. "At higher light intensities, we expect a larger pool of photosynthetically derived NADPH inside the cell," Do you have direct evidence (either your own, or from a reference) to support this? Photosynthesis is pretty tightly regulated, so my expectation would be that as soon as NADPH levels start to creep up, photons are shunted to non-photochemical quenching instead of to the photosystems and NADP reduction. In which case, NADPH levels might not depend on light levels. There should be papers about this in the biochemical literature.

Page 9, lines 5-10. Larger pool of reduced NADPH could also mean a longer lifetime, and greater D/H exchange.

Table 1. Can you at least include the initial alkalinity and/or pH for the high-light experiments? It is not essential, just seems weird not to report them given the emphasis on that variable of the rest of the paper.

---

## Referee Comment (RC3) · Anonymous Referee #3 · 26 Sep 2017

The study "Effects of alkalinity and salinity at low and high light intensity on hydrogen isotope fractionation of long-chain alkenones produced by Emiliania huxleyi" by Gabriella M. Weiss, Eva Y. Pfannerstill, Stefan Schouten, Jaap S. Sinninghe Damsteİ, and Marcel T.J. van der Meer is an important step forward in the quest to understand the environmental sensitivities of hydrogen isotope fractionation during lipid biosynthesis in unicellular photoautotrophs. The experimental design, measurements, analysis and interpretations are all high quality, and I have no major criticisms of the work. However, I do have a few suggestions about ways that the manuscript could be improved,

which I have outlined below. I recommend publication once the authors have had a chance to consider these, and the input from the other reviewers.

General comments:

I found the discussion on mechanisms of hydrogen isotope fractionation to be well thought out and referenced, but I think that a slightly expanded discussion on the growth conditions that E.hux experiences in the environment, seasonality of lipid production, and effects of growth rate, light, and nutrients, etc. on alkenone production might be helpful to guide the discussion on alkenone H-isotope fractionation. This is a very well-studied organism after all, and we benefit from decades of research on these factors due to the work that has been done for understanding Uk37 and 13C/pCO2 applications. I suspect that these lessons could be applied to the present work a bit more than they are currently.

In general the paper is careful to specify when discussing alkenones produced by E.Hux from those produced by other haptophytes, but there are a few cases where this isn't clear and I'd recommend clarifying these.

The quality of the writing is fine for the most part, although I do have a few suggestions and typos that I've outlined below, so I would also just recommend here that it gets read again with this in mind.

Detailed comments:

P.2L1-10 - I would not refer to continental bodies of water as meteoric this way. Meteoric implies precipitation-derived and that dD = 8*d18O +10, and many continental bodies of water are enriched due to evaporation, making them non-meteoric. Also, meteoric lakes and rivers are very fresh, making the statement about "low salinity" a little odd. I might rephrase this sentence to say something like "Therefore, most lakes and rivers that are fed by precipitation (i.e. meteoric waters) are characterized by a depleted isotopic signature. As these waters drain into the ocean and mix with seawater the

result is to lower both the sea surface salinity as well as the water isotope value, as also occurs during direct precipitation on the ocean."

P2.L17 - It is a little confusing here to use alpha without really defining it, especially here in this context since the sentence describes hydrogen isotope fractionation decreasing with increasing salinity, meaning an increase in the fractionation factor in this case. I think you don't need to define alpha right here anyway, so I might suggest leaving it until later, at the first actual required use. When the first use and definition do come, I'd also recommend including an equation at that point in line with the normal text because the definition of alpha is always application specific depending on the nature of the isotope system and product/substrate applications in any particular study.

P2.L21-27 – Leduc et al., EPSL, 2013 would be another reference that might be worth discussing here in example applications.

General - Throughout the entire manuscript, delta values (of all types) would be better used in the text with the word "value" (e.g. "d18Oforam values from the same region. . ..", as opposed to "d18Oforam from the same region. . ..")

P2. – Acronyms should be defined at first use or not used in my opinion, so on this page "LGM" and "SPM".

P2-3 – This paragraph might cite Nelson and Sachs, OG, 2014 in the discussion of field studies, and Wolhowe et al., Biogeosciences, 2009, in the discussion of growth phase. Also be sure to be clear about which observations/studies apply specifically to E.Hux and which don't.

P3.L7 – Reference needed for "the culture experiments"

P3.L9 – change "on" to "by"

P3.L17 – Reference needed for "most of the culture studies"

P3.L29 – Not necessarily here, but somewhere in the paper there should be a discussion about why a non-calcifying strain was selected.

P4.L1-5 – Somewhere in here it would be good to state the pH of the cultures too.

P5.L.19 - change "and therefore" to "and were therefore"

P5.L20 - provide reference when making a comparison to "previous studies"

P5.L21 - remind us here, as well as in the caption for figure 1, why those dDwater values are so high in that one group of samples. Maybe different colors for the modified alkalinity samples in the figure?

P5.L24 - alpha values should also be discussed in the text using the word "value" so change "a37" to "a37 values".

Section 3 - Results sound better when consistently described in the past tense in my opinion (e.g. P5.L24-25 as: "A strong linear relationship between $\alpha$C37 values and salinity was observed in both experiments). Either way, be consistent about tense use throughout.

P6.L10-15 - It would be useful here to provide a reference to what surface ocean light levels are and how these dissipate with depth.

P6.L15 - "statistically similar" – be quantitative

P6.L23-25 - Which individual C37 alkenone from the Sachs16 reference is being used to compare to the mixed C37 alkenone dD values reported in the other studies?

P6.L24 - Chivall14 used a coastal producer, no? The equation is also not listed in table 2. Should this reference be omitted from this list?

P6.L30 - I think that this issue of production depth/light exposure could benefit from a slightly expanded summary and literature survey. This gets to one of my general comments at the beginning. It might also be useful to comment on where in the ocean one might expect to find light levels that might cause a large H-isotope effect (i.e. <

~100 umol, based on the van der Meer, GCA, 2015 paper).

Section 4.2 - Describe the statistical similarities in slopes and differences in intercepts in quantitative terms. What thresholds were applied?

P7.L6 - change "due differences" to "due to differences"

P7.L6 - As written, "different sources of hydrogen" is probably not the best language. I gather that the implication invokes biochemical mechanisms relating to the routing of hydrogen during biosynthesis, but the way it is currently makes it sound like they are accessing different source water, which is probably not the intention.

P7.L5-9 - what about chemostats vs. batch cultures? That probably deserves a mention somewhere in here in comparing to Sachs16.

P7.L13-15 - Nelson and Sachs, GCA, 2014 would be worth including in this list of references

P7.L14 – I would specify "algal" or "unicellular" photoautotrophs, or include references to alpha-salinity relationships in plants (Aichner et al., OG, 2017; Ladd and Sachs, OG, 2012).

P8.L22 - change "by OPP" to "by the OPP"

Figure 1 - I suggest labeling the panels directly on the graphs to allow them to be read without looking at the caption……. I also would personally prefer if the graphs were the same width, and were aligned with each other. I'd also use the same x-axis scale for both, and would note the y-axis scale differences in the caption…. State clearly that the gray shaded areas are confidence intervals (they are, right?) and provide the threshold that was used to define these in the caption

Figure 2 - The font in the axis labels, as well as the plotted symbols look like they were compressed vertically. Can this be fixed so they don't look squished like this? …….. label the experimental design directly on panels a and b, or consider adding

[Figure]

this information using a legend to indicate symbol color. . . . . . see my comment about confidence intervals as related to figure 1 above.

Figure 3 - See my comment about confidence intervals as related to figure 1 above.

[Figure]

---

## Author Comment (AC2) · 6 Oct 2017

We would like to thank Dr. Sessions for the comments on our manuscript, which we will take into consideration and would like to address as "Response:" following the original comment.

My only general comment is to question why the authors chose to describe culture conditions in terms of alkalinity rather than pH. With [DIC] fixed by equilibrium with atmospheric PCO2, alkalinity and pH are in a sense interchangeable (fixing one uniquely

determines the other). Thus the same experiments could be described in terms of either parameter. Alkalinity is probably more popular among oceanographers, but pH is much more widely used among biologists. And I might argue that there is some reason to think that cellular H isotope fractionation depends more on the concentration of H+ (i.e., pH) than on the ability to consume H+ (i.e., alkalinity). So my suggestion is to at least consider describing the first series of experiments as a pH series, rather than an alkalinity series. Or maybe there is a way to gracefully do both.

Response: We understand that in the natural environment alkalinity and pH are linked, however, in our experiment, we kept pH constant (7.9 $\pm$ 0.07) and only changed the alkalinity by adding NaHCO3 and Na2CO3 to increase and concentrated HCl to decrease total alkalinity in our original media.

I was curious why a non-calcifying strain of E. hux was chosen. Perhaps it simplifies controlling alkalinity? In any case, it would be worth a few sentences of explanation about why you chose this strain, and how it might relate to strains that are prevalent in the oceans. Is it likely to be representative of strains that produce alkenones in most marine sediments?

Response: A non-calcifying strain was chosen because we wanted to avoid changes in the total alkalinity of the media caused by the organisms themselves, which has been shown to be the case in the natural environment by Hooligan et al., 1993 (Global Biogeochemical Cycles 7). We agree that a calcifying strain might be more representative of the natural environment, but in order to get a handle on whether alkalinity has an effect on hydrogen isotope fractionation or not, we chose a non-calcifying strain that has been used in previous studies (M'Boule et al., 2014). Furthermore, all previous culture experiments were done with non-calcifying strains, therefore, by using a non-calcifying strain we would better compare our results with previous results. We will add this to a revised version of the manuscript.

Section 2.1. Please tell us how you measured (or calculated) alkalinity?

Response: Alkalinity was measured by titration with 0.1 M HCl, and calculated using Gran plots. (G. Gran: Determination of the equivalence point in potentiometric titrations. Part II, Analyst 1952, 77, 661). We will add this to the manuscript.

Page 6, line 25. You say that you performed a statistical comparison, and then that "This showed a strong similarity between slopes..". What does strong similarity mean in statistical terms? They are indistinguishable? Given that the slopes differ between experiments by more than a factor of 2, this is probably more a statement about variability between experiments rather than a constant slope. Seems like the discussion of this 'similarity' could be a bit more nuanced. Differences of a factor of _2 would still make a huge difference in reconstructing seawater salinity, even if they are statistically indistinguishable.

Response: True. We were referring to the fact that the slopes were not statistically different because we wanted to examine the data as a whole data set, but we realize that this might not actually be useful for reconstructing salinity because of the differences mentioned above. Furthermore, we believe that the individual experiments themselves (i.e., conducted by different people in different labs using different techniques) do play a large role in the observed differences between slopes, as well as different strains of E. huxleyi that were utilized.

Page 7, line 6. The differences in intercepts amount to a range of nearly 78‰Ź That does not seem (to me, at least) plausible to explain solely by interlaboratory differences. Maybe modify the text to say that "part of" the differences could possibly be attributed to this.

Response: We will fix this.

Page 7, lines 20-35. The term "photosynthetically-derived NADPH" struck me as a little odd, especially in contrast to the more precise "pentose-phosphate pathway". Photosynthesis both produces (in photosystem I of the light reactions) and consumes (in CO2 fixation of the Calvin cycle reactions) NADPH. It would thus be more precise to

refer to NADPH from the "light reactions of photosynthesis", or to "ferredoxin-NADP+ reductase (FNR) in photosystem 1", etc.

Response: We will fix this.

Page 7, line 30-33. I like this explanation, a lot. It is the best one I have heard yet.

Response: Thanks.

Page 9, line 4. "At higher light intensities, we expect a larger pool of photosynthetically derived NADPH inside the cell," Do you have direct evidence (either your own, or from a reference) to support this? Photosynthesis is pretty tightly regulated, so my expectation would be that as soon as NADPH levels start to creep up, photons are shunted to nonphotochemical quenching instead of to the photosystems and NADP reduction. In which case, NADPH levels might not depend on light levels. There should be papers about this in the biochemical literature.

Response: Due to balancing between ATP and NADPH production and consumption within the cell (Walker et al., 2014, Plant Physiology 165), NADPH formation dominates at high light levels, whereas ATP synthesis dominates at lower light levels (Beardall et al., 2003 in Photosynthesis in Algae), leading to the idea of a larger pool of photosynthetically derived NADPH inside the cell. We will add these references to our discussion of this topic.

Page 9, lines 5-10. Larger pool of reduced NADPH could also mean a longer lifetime, and greater D/H exchange.

Response: Yes, true. We agree.

Table 1. Can you at least include the initial alkalinity and/or pH for the high-light experiments? It is not essential, just seems weird not to report them given the emphasis on that variable of the rest of the paper.

Response: Alkalinity and pH were not measured for the high light experiments, since

they were performed separately and not originally considered to be discussed along-side the alkalinity/salinity experiment results, but we used filtered North Sea water, so the alkalinity is presumably around 2.3 (Brasse et al., 1999. Journal of Sea Research 42.)

———————————————————

---

## Author Comment (AC3) · 6 Oct 2017

Anonymous Referee #3 The study "Effects of alkalinity and salinity at low and high light intensity on hydrogen isotope fractionation of long-chain alkenones produced by Emiliania huxleyi" by Gabriella M.Weiss, Eva Y. Pfannerstill, Stefan Schouten, Jaap S. Sinninghe DamsteÌAËŻ, and Marcel T.J. van der Meer is an important step forward in the quest to understand the environmental sensitivities of hydrogen isotope fractionation during lipid biosynthesis in unicellular photoautotrophs. The experimental design, measurements, analysis and interpretations are all high quality, and I have no major criticisms of the work. However, I do have a few suggestions about ways that the manuscript could be improved, which I have outlined below. I recommend publication once the authors have had a chance to consider these, and the input from the other reviewers.

We would like to thank Anonymous Referee #3 for the constructive comments, which we will take into consideration and would like to address below as "Response:" following the original comment.

General comments: I found the discussion on mechanisms of hydrogen isotope fractionation to be well thought out and referenced, but I think that a slightly expanded discussion on the growth conditions that E.hux experiences in the environment, seasonality of lipid production, and effects of growth rate, light, and nutrients, etc. on alkenone production might be helpful to guide the discussion on alkenone H-isotope fractionation. This is a very well-studied organism after all, and we benefit from decades of research on these factors due to the work that has been done for understanding Uk37 and 13C/pCO2 applications. I suspect that these lessons could be applied to the present work a bit more than they are currently.

Response: Noted, we will incorporate them into our discussion.

In general the paper is careful to specify when discussing alkenones produced by E.Hux from those produced by other haptophytes, but there are a few cases where this isn't clear and I'd recommend clarifying these.

Response: We will clarify these.

The quality of the writing is fine for the most part, although I do have a few suggestions and typos that I've outlined below, so I would also just recommend here that it gets read again with this in mind. Detailed comments: P.2L1-10 - I would not refer to continental bodies of water as meteoric this way. Meteoric implies precipitation-derived and that dD = 8*d18O +10, and many continental bodies of water are enriched due to

evaporation, making them non-meteoric. Also, meteoric lakes and rivers are very fresh, making the statement about "low salinity" a little odd. I might rephrase this sentence to say something like "Therefore, most lakes and rivers that are fed by precipitation (i.e. meteoric waters) are characterized by a depleted isotopic signature. As these waters drain into the ocean and mix with seawater the result is to lower both the sea surface salinity as well as the water isotope value, as also occurs during direct precipitation on the ocean."

Response: Thank you for the suggestion, we will rephrase in the revised manuscript.

P2.L17 - It is a little confusing here to use alpha without really defining it, especially here in this context since the sentence describes hydrogen isotope fractionation decreasing with increasing salinity, meaning an increase in the fractionation factor in this case. I think you don't need to define alpha right here anyway, so I might suggest leaving it until later, at the first actual required use. When the first use and definition do come, I'd also recommend including an equation at that point in line with the normal text because the definition of alpha is always application specific depending on the nature of the isotope system and product/substrate applications in any particular study.

Response: Noted, we will take this into account in a revised version.

P2.L21-27 – Leduc et al., EPSL, 2013 would be another reference that might be worth discussing here in example applications.

Response: Noted, we will include Leduc et al., 2013 in this discussion.

General - Throughout the entire manuscript, delta values (of all types) would be better used in the text with the word "value" (e.g. "d18Oforam values from the same region. . ..", as opposed to "d18Oforam from the same region. . ..")

Response: Noted and will be changed.

P2. – Acronyms should be defined at first use or not used in my opinion, so on this page "LGM" and "SPM".

Response: Noted and will be changed.

P2-3 – This paragraph might cite Nelson and Sachs, OG, 2014 in the discussion of field studies, and Wolhowe et al., Biogeosciences, 2009, in the discussion of growth phase. Also be sure to be clear about which observations/studies apply specifically to E.Hux and which don't.

Response: Noted and will include both references in the discussion as well as better clarify which species we are referring to.

P3.L7 – Reference needed for "the culture experiments"

Response: Will correct.

P3.L9 – change "on" to "by"

Response: Will correct.

P3.L17 – Reference needed for "most of the culture studies"

Response: Will correct.

P3.L29 – Not necessarily here, but somewhere in the paper there should be a discussion about why a non-calcifying strain was selected.

Response: Noted, see comment to Referee A. Sessions.

P4.L1-5 – Somewhere in here it would be good to state the pH of the cultures too.

Response: Noted, see comment to Referee A. Sessions.

P5.L.19 - change "and therefore" to "and were therefore"

Response: Will correct.

P5.L20 - provide reference when making a comparison to "previous studies"

Response: Will correct.

P5.L21 - remind us here, as well as in the caption for figure 1, why those dDwater values are so high in that one group of samples. Maybe different colors for the modified alkalinity samples in the figure?

Response: Will correct.

P5.L24 - alpha values should also be discussed in the text using the word "value" so change "a37" to "a37 values".

Response: Will correct.

Section 3 - Results sound better when consistently described in the past tense in my opinion (e.g. P5.L24-25 as: "A strong linear relationship between _C37 values and salinity was observed in both experiments). Either way, be consistent about tense use throughout.

Response: Will correct.

P6.L10-15 - It would be useful here to provide a reference to what surface ocean light levels are and how these dissipate with depth.

Response: Will correct.

P6.L15 - "statistically similar" – be quantitative

Response: Will correct.

P6.L23-25 - Which individual C37 alkenone from the Sachs16 reference is being used to compare to the mixed C37 alkenone dD values reported in the other studies?

Response: We used a weighted mean average to compare the separated C37 alkenones measured by Sachs et al., 2016 to the integrated alkenones in our experiments and the previous experiments of E. huxleyi.

P6.L24 - Chivall14 used a coastal producer, no? The equation is also not listed in table 2. Should this reference be omitted from this list?

Response: Yes. Will correct.

P6.L30 - I think that this issue of production depth/light exposure could benefit from a slightly expanded summary and literature survey. This gets to one of my general comments at the beginning. It might also be useful to comment on where in the ocean one might expect to find light levels that might cause a large H-isotope effect (i.e. <100 umol, based on the van der Meer, GCA, 2015 paper).

Response: Will correct. See comment to Anonymous Referee #1.

Section 4.2 - Describe the statistical similarities in slopes and differences in intercepts in quantitative terms. What thresholds were applied? Response: Will correct.

P7.L6 - change "due differences" to "due to differences"

Response: Will correct.

P7.L6 - As written, "different sources of hydrogen" is probably not the best language. I gather that the implication invokes biochemical mechanisms relating to the routing of hydrogen during biosynthesis, but the way it is currently makes it sound like they are accessing different source water, which is probably not the intention.

Response: Will correct.

P7.L5-9 - what about chemostats vs. batch cultures? That probably deserves a mention somewhere in here in comparing to Sachs16.

Response: Will correct.

P7.L13-15 - Nelson and Sachs, GCA, 2014 would be worth including in this list of references

Response: Will correct.

P7.L14 – I would specify "algal" or "unicellular" photoautotrophs, or include references to alpha-salinity relationships in plants (Aichner et al., OG, 2017; Ladd and Sachs, OG,

2012).

Response: Will add these references.

P8.L22 - change "by OPP" to "by the OPP"

Response: Will correct.

Figure 1 - I suggest labeling the panels directly on the graphs to allow them to be read without looking at the caption. . .. . .. I also would personally prefer if the graphs were the same width, and were aligned with each other. I'd also use the same x-axis scale for both, and would note the y-axis scale differences in the caption. . .. State clearly that the gray shaded areas are confidence intervals (they are, right?) and provide the threshold that was used to define these in the caption Figure 2 - The font in the axis labels, as well as the plotted symbols look like they were compressed vertically. Can this be fixed so they don't look squished like this? . . .. . ... label the experimental design directly on panels a and b, or consider adding this information using a legend to indicate symbol color. . .. . . see my comment about confidence intervals as related to figure 1 above. Figure 3 - See my comment about confidence intervals as related to figure 1 above.

Response: We will fix these issues with the figures.

---

## Referee Comment (RC4) · A. Sessions (Referee) · 11 Oct 2017

In regard to the discussion about whether pH and DIC change in the alkalinity experiments:

OK, your response makes sense - I had not read carefully enough to realize that pH was going to stay constant in the alkalinity series.

It is worth noting though, that by decreasing/increasing alkalinity at constant pH you are also changing [DIC]. Its not clear whether or not the latter parameter has any effect

on D/H fractionation, but it is at least plausible.

To help make all this clear, I think it would be helpful to add a sentence on page 4, line 7, that says something to the effect of: In all the alkalinity experiments, pH remained roughly constant at 8.5-8.7; forced equilibration with atmospheric $CO_2$ under these conditions means that [DIC] also changed by a factor of $\sim$4.

Alex Sessions

---

## Referee Comment (RC5) · Anonymous Referee #4 · 17 Oct 2017

The paper by Weiss et al. presents new data from a laboratory experiment aiming to clarify whether and how strong salinity and light intensity affect the hydrogen isotope fractionation during alkenone biosynthesis. Such results pave the way towards an application of algal lipid biomarker hydrogen isotope ratios as a paleosalinity proxy. While similar experiments have been conducted before and salinity and light intensity have been found to affect the hydrogen isotope fractionation, results from the current study test in particular the effect of alkalinity (which can change independently of salinity) on the isotope fractionation. It therefore adds to the understanding of how representative

the previous findings from laboratory cultures are for the natural environment. The study finds that alkalinity does not affect the isotope fractionation and finds similar relationships between isotope fractionation and salinity as observed in previous studies. They also find that changes in light intensity do not change the relationship between salinity and isotope fractionation. These results provide a more robust base to use alkenone D/H ratios as a paleosalinity proxy and may therefore help to identify the actual cellular mechanism responsible for the observed changes in fractionation. While not representing groundbreaking new insights, the study adds to the growing body of literature on this subject. The study is well designed and interpretations are supported by the data. I believe this study should be published after some minor changes. In particular I suggest some clarification of statistical data treatment and a few more detailed descriptions of the experimental setup.

General comments:

In the study a non calcifying strain of e.hux was used. The authors discuss this to some degree, but a bit more detailed discussion, on how representative these results would be for the natural marine environment, where mostly calcifying strains produce the alkenones, should be part of the discussion.

It appears that the statistical data treatment was done using the three replicate data points as individual datapoints – I think it would make more sense to calculate the mean of the replicates and present the standard error of the mean for each treatment. This applies to the actual slope and intercept calculations as well as for the figures and the estimation of the error of the actual regressions (i.e. the shaded area around the regression lines in the figures), see also below.

The figures could need some more explanation, in the text but also the figure captions. See detailed comments below.

Detailed comments:

P6 line 30-31: Can you separate this sentence into 2? It conveys important information, but sounds a bit awkward.

P7 line 3-4: Can you mention by how much the intercepts from the other studies vary? I believe it would be instructive to present the data from the current study and previous data from the literature in one graph, see comment below (Table 2).

P7 line 9: Header for this section does only mention salinity but the second half of the paragraph deals with light intensity. Either separate the paragraph into 2 or mention light intensity in the headline.

P7 line 14: In the cited studies not only alkenones, fatty acids and sterols were analyzed, also alkanes and isoprenoids if I remember correctly. I think it would be important to mention that in all these compound classes similar salinity effects have been observed. This is important to identify the underlying mechanism.

P8 line 20-21: Interesting hypothesis. Would this hold some advantage for the cell, i.e. using more OPP derived NADPH under higher salinity? Or could this be the result of less water exchange (extracellular with intracellular)?

Page 8 in general: This is a good summary of the hypotheses being discussed for the observed salinity-fractionation relationship. Except a few points (see above) these have all been proposed in previous papers which have identified the salinity-fractionation dependency. This could be mentioned more explicitly. I suggest to give credit to these papers here, for example in the section about osmolytes the first papers proposing this idea as a factor for the observed change in fractionation, should be cited.

Figure 1a: Can you briefly explain, why the culture media water dD values at salinity of 35 are so different from the rest?

Figure 1b: I suggest to use the same scale on the x and y axis as in a)

Figure 2: also here I suggest to use the same scaling of the x and y axis (at least for salinity). I think that statistically it would make more sense to use the mean of the

[Figure]

replicates and their standard deviation for the plots and also to estimate the error of the regression line (standard error of the mean).

Figure 2c: Can you briefly explain the alpha variability at an alkalinity of 2.5?

Figure 3: also here, I suggest to sue the same axis scaling (both for alpha and growth rate and salinity). Clearly, and this is the main point of the paper, salinity has a much stronger effect on isotope fractionation compared to growth rate and this would be easily visible in the graphs, when the same axis scaling is used. Also, if a regression line is plotted through the data, you imply a statistically significant correlation. Is that so in all cases, and if so, then you should present the statistical parameters (p value). If it is not statistically significant, no line should be plotted through the data.

Table 2: I think it would be useful to see these data compared to the data from the current study in a graph.

---

## Author Response (AR1)

Weiss et al. (2017) conducted two experiments using batch cultures of the haptophyte Emiliania huxleyii (strain CMP1516) to determine 1) how alkalinity (separate from salinity) might impact hydrogen isotope ratios in alkenones and 2) if high light conditions influence the previously reported salinity-fractionation relationship in alkenones. The results indicate that the alkenone hydrogen isotope salinity-fractionation relationship is robust (and similar to previously reported relationships) regardless of alkalinity and light level, which are useful and interesting findings. I recommend this manuscript for publication but request that the authors address the following issues: 1) more information about natural light levels, 2) separate alkalinity as the sole variable as part of the analysis, 3) improve the discussion about mechanisms (if possible include additional lipid isotope and lipid concentration data to support the lack of light effect), plus a handful of technical issues listed below.

*We would like to thank the anonymous referee #1 for the constructive review of our manuscript. We will address the comments below in italics following the original comments.*

High light growth conditions in the world's oceans: The idea that most alkenones are produced in the high light of surface waters is mentioned in several places (abstract line 5; intro p.3 line19; discussion section 4.2 line 30; conclusion line 26), but without references to support this claim. As satellite imagery clearly illustrates, major coccolithophore blooms occur in highly productive (mainly coastal) areas that are seasonally growth-limited and along the equator, and presumably bloom alkenones are produced in high light conditions near the surface (but surely also deeper in the water column where light is limited due to self-shading from bloom turbidity?). As is stands, the manuscript just states that high light alkenone production likely predominates. But to strengthen your argument and support your experimental design and data interpretation, the predominance of high light alkenone production should be illustrated with references that indicate

1) alkenones are mainly produced in high light at the surface,

2) how ocean surface water light level ranges compare to light ranges in your study,

3) how surface bloom productivity compares to non-bloom conditions/areas, and

4) that these "high light" bloom alkenones are indeed exported to the sediments (more so than "low light" alkenones). Because in vast areas of the ocean, it seems the primary source of alkenones comes from fairly deep in the water column where light levels are a fraction of surface values. For instance, this has been demonstrated at BATS see Fig 1 in Krumhardt et al. 2016 (doi:10.5194/bg-13-1163-2016), at ALOHA Table 3 Prahl et al. 2005 (doi:10.1016/j.dsr.2004.12.001) and further afield Ohkouchi et al. 1999 (DOI:¢a10.1029/1998GB900024).

The vast areas of the ocean with potential deepwater/lowlight alkenone production should be addressed/acknowledged with clarification of how alkenones produced in non-saturating light conditions do or do not affect the alkenone dD paleosalinity proxy in some regions.

The same references used in van der Meer (2015) (in quoted text below) are helpful, and should again be used here – and if possible, supplemented with additional references: "E. huxleyi, for instance, is thought to thrive under high light conditions, at mixed layer depths generally <30 meter (Tyrrell and Merico, 2004; Harris et al., 2005). They outcompete other algal species that suffer from photoinhibition under these conditions, a process that is apparently absent in E. huxleyi (Nanninga and Tyrrell, 1996)."

*Response: Yes, we agree that this would be a useful addition, and would add that although* Krumhardt et al., 2016 *do indicate that haptophytes are abundant below surface water layers, they also explain that haptophyte indicative pigments were abundant in the upper 30m, especially during spring as well as the observation that during the mid-90s and over the last 6 years of the data set, that "Chl $a_{hapto}$ was more concentrated especially in the upper 30m of the water column", allowing us to infer that deep-dwelling haptophytes are perhaps not the most dominant. We also would like to point out that these blooms are not only occurring along the equator, but also occur in the high latitudes (Holligan et al., 1993). Furthermore, based on UK'37 core-top calibration, we can be confident that alkenones preserved in the sediments are largely reflecting surface water temperatures during the time of the year that haptophytes are known to bloom (Müller et al., 1998). Additionally, ocean surface water light levels span a range from zero to over 800 PAR (over 1600 μmol photons $m^{-2}$ $s^{-1}$) (Frouin and Murakami, 2007), and haptophytes are thought primarily to bloom at light intensities above 500 μmol photons $m^{-2}$ $s^{-1}$ (Nanninga and Tyrrell, 1996), so we feel that the light intensity used in our study accurately represents environmental conditions. Most culture studies to date have been performed at light intensities much lower than 600, with Schouten et al., 2006 being one of the higher light intensity studies at 300 μmol photons $m^{-2}$ $s^{-1}$. Culture studies already cover the low light range. Van der Meer et al., 2015 suggested that at light intensities above 200 μmol photons $m^{-2}$ $s^{-1}$ α responds differently to changes*

*in light intensity than below. This and the observation that haptophytes tend to bloom at light intensities above 500 μmol photons m$^{-2}$ s$^{-1}$ warrants the study of hydrogen isotope fractionation in response to salinity at high light intensity. We have incorporated this discussion throughout the revised manuscript.*

Alkalinity: Fig 2c clearly shows that there is not a relationship between total alkalinity and fractionation (alpha). However, the discussion of the alkalinity results in the low light treatments is not entirely satisfying (p.6 lines 3-8), and it can be confusing when plots include data with many changing variables that are known to influence the isotopic composition of lipids. It would be useful (even if just in a supplement) to break up the different environmental parameters. Looking only at the low light cultures grown at salinity _35 (12 cultures, salinity range 34.5-35.4, alkalinity range 1.39-4.58, growth rate range 0.69-0.93, alpha range 0.795-0.806) (ie, excluding the cultures that have salinities of _26, _31, _37, or _42, nearly constant alkalinity _2.4, but growth rate range of 0.65-0.93, and alpha range of 0.776-0.824) . . . It is noteworthy that for these 12 cultures there is a correlation between total alkalinity and specific growth rate (=0.05(0.01) * AT + 0.676(0.04), R2 = 0.51, p = 0.0056) in addition to a correlation between specific growth rate and fractionation (= 0.03(0.01) * + 0.776(0.01), R2 = 0.47, p= 0.0081) (figure attached as example). Although this is a very small range in growth rate (_.7 to .9) compare to a previous study of the effect of growth rate on alkenones using a combination of chemostats and continuous cultures (Sachs and Kawka, 2015), it is striking (and probably noteworthy) a) how well they are correlated and b) that it is in the opposite sense of previous 2015 study. In the end, there is still not a significant
relationship between alpha and total alkalinity – could it be that the minor growth rate effect here is overwriting any alkalinity effect? It is probable that you already considered all of this and decided it didn't fit in the paper – but it would be useful for the interested reader to be able to reference in at least the supplement.

*Response: We agree that there appears to be a correlation between specific growth rate and total alkalinity for the batch cultures grown at constant salinity over a range of alkalinity. It cannot entirely be ruled out that the minor growth rate effect could be overwriting a potential alkalinity effect, however, as the reviewer has already remarked, the growth rate-α relationship is a positive one, which is opposite to what has been described previously, meaning that the alkalinity effect would have to be negative to be counteracted by growth rate in this case. Since salinity and alkalinity are usually positively correlated, this would suggest that alkalinity would also counteract the salinity effect on α. We and Sachs et al., 2016 do not observe stronger positive correlations (steeper slope) when alkalinity is removed as a variable between α and salinity, which is what would be expected. Looking at the plots and statistical data generously provided by the reviewer, we feel that the correlations are barely statistically relevant, especially according to Johnson et al. (2013) who suggest a cut-off P-value of 0.005 for truly significant findings.*

Mechanisms: It is welcome that some effort is put forth to explain the cellular mechanisms of the salinity response at both high and low light. However, this part of the discussion seems confused and needs some work by improving the organization, offering introductory or concluding summaries to help identify your main points, supporting hypotheses with literature, perhaps a schematic outlining your favorite mechanism (which one is most likely here and why?), and the issues below. A little bit of reorganization could help since it seems like the discussion tries to deal with the high light and low light salinity response in both section 4.3 and 4.4. One way to improve this is if section 4.3 can just concentrated on salinity (regardless of light), then address why a different response at high light was expected, and then finally address why there was not a strong light effect here (but more on this below). The first paragraph of section 4.3 is very long, and the main point of it is lost in the length. Maybe at the start let the reader know how many mechanisms you will cover, then at the end summarize which one seems most likely. Can you rule any out with the results from your experiment? A lot of attention is given to metabolically reduced NADPH, such as that generated through OPP. However, there are a few things to consider. NADPH can't cross organelle membranes – so p.8 lines 17-21 doesn't make sense, unless the complete OPP pathway (and final steps to the alkenone synthesis pathway?) were present in this hypothetical closed compartment.

*Response: We have reorganized and combined section 4.3 and 4.4 to be clearer and more focused, under the heading, "4.3 Potential mechanisms for salinity and light responses". We were referring to the final steps of the alkenone synthesis pathway, assuming alkenones are synthesized from fatty acids and short chain fatty acids are coming from the chloroplast and are elongated somewhere else with a potential change in NADPH source, possibly the OPP pathway. Since it has been proven difficult to measure hydrogen isotopic composition of some of the key players in biosynthesis with high enough accuracy (intracellular water, NADPH from different sources, etc.), the idea behind this discussion is to try and combine all the information we do have and try to reason what the possible mechanism behind the salinity effect might be.*

Alternatively you might invoke the import of alkenone precursors (fatty acids? Pyruvate?) used to build alkenones that reflect proportional changes in photosynthetically vs metabolically reduced NADPH pools (and would have to introduce them around p.7 line18).

*Response: Noted, alkenone precursors could be the reason that OPP-derived NADPH might end up in alkenones, meaning that OPP-derived NADPH might be more of an influence for alkenones than photosynthetically derived NADPH.*

Lack of light effect: p.9 line3 states "we do not see a clear relationship between hydrogen isotope fractionation and light intensity." But there is no mention of this lack of relationship in the results, where should the reader go to visualize this lack of relationship? Do you mean between the 24 low light cultures compare to the 14 high light cultures in this study? There aren't really cultures in the high light with the same growth rate and salinity as the cultures in the low light (although 3 HL cultures at S=35.9 and 12 LL cultures at S=34.5-35.4 actually do show a (very small) significant increase in alpha at the higher light level – however, that could in theory be due to the slightly saltier HL media or the slightly lower HL growth rate rather than the higher light). More importantly, I am not convinced that there is no light effect with only 2 light levels (75 and 600), especially considering the previously observed non-linear relationship between high light in strain RCC1238 and the fact that different strains have shown different responses (van der Meer et al. 2015). Is it possible that both the 75 and 600 light levels are above the range where a response would be detected for this strain? After all, growth rate is lower in the higher light cultures. Even though (it seems) the goal of the paper is not to characterize the light effect (but to test if the salinity effect holds at high light), this part of the discussion should consider the option that if batch cultures were grown at additional light levels, then a light effect in this strain might be apparent.

*Response: Based on van der Meer et al., 2015, we expected a significant decrease in fractionation between our high light experiment and those performed at low light intensities. However, the α-salinity relationship at high light intensities plots on top of all of the other experiments. While we do understand that our results cannot necessarily be extrapolated to all strains, we argue that a statistically similar α-salinity relationship is observed independent of all the other parameters that were being tested in previous experiments, specifically with regards to differences between low (this study, M'Boule et al., 2014), intermediate (Schouten et al., 2006; Sachs et al., 2016) and high light (this study) intensities.*

p.9 line 4 – "At higher light intensities, we expect a larger pool of photosynthetically derived NADPH inside the cell" – yes of course, but another thing to consider is this doesn't necessarily mean that this NADPH is available for alkenone synthesis.

*Response: Yes, we consider and discuss this (p9 lines 12-15)*

At high light levels cells might be working hard to dump this extra reductive power (into alkenones? or other molecules types – is there literature on algae at different light levels you can turn to?), along with reducing light harvesting capacity, which might be a reason for the lack of light effect (if there is truly a lack of light effect).

*Response: There is literature about how other types of photoautotrophic microorganisms deal with different light levels, for example, it has been observed in* Synechococcus *growing in microbial mats that in order to avoid light, they try to decrease the surface area exposed to light by "standing upright", thereby reducing their surface area. (Ramsing et al., 2000). However, haptophytes, specifically* E. huxleyi*, are not believed to be photoinhibited organisms, especially not at the light levels we are discussing (Nanninga and Tyrrell, 1996). Van der Meer et al., 2015 did show that there is a light effect on fractionation between the range of 60-600μmol photons m$^{-2}$ s$^{-1}$, however, here we are focusing on the α-salinity relationship under high light conditions and how that compares with previous experiments under lower light intensities.*

It seems like cells have many different options for dealing with high light situations. Were alkenone (and other lipid) concentrations measured? Were there any large concentration differences at different light levels indicating strategies for dumping excess NADPH? p.9 line 9 – why do you think that transhydrogenase activity is increased at high light?

*Response: No, we did not compare the concentrations at high and low light conditions. We think transhydrogenase activity is increased at high light because excess reducing power is harmful to the cell and there is a good chance the algae will try to get rid of it either by biosynthesizing storage products or turning it into a less harmful product.*

Can you provide a reference to demonstrate that this is a response to high light in haptophytes or even just eukaryotic (or even prokaryotic) algae? Rokitta et al. (2012) doi:10.1371/journal.pone.0052212 might be helpful here.

*Revised to: "Ocean surface light levels span a range from zero to over 800 PAR (over 1600 µmol photons m$^{-2}$ s$^{-1}$) (Frouin and Murakami, 2007), and haptophytes not believed to be photoinhibited and primarily bloom at light intensities above 500 µmol photons m$^{-2}$ s$^{-1}$ (Nanninga and Tyrrell, 1996). Furthermore, E. huxleyi has been shown to adapt to different light conditions by expressing different genes under high and low light conditions (Rokitta et al., 2012)."*

Regarding the last paragraph of section 4.4 – is there any chance hydrogen isotopes were measured on other lipids from these cultures (fatty acids, brassicasterol, phytol. . .except probably not phytol if you didn't saponify)? This information could be extremely helpful for illuminating cellular changes in response to environmental variables, and the interplay between different pools of cellular hydrogen, as it was for Sachs et al. (2017) that found different responses to different fatty acids, phytol, and a sterol, in a diatom grown under constant salinity, temperature, and growth rate at a range of light levels. In other words, just because there wasn't a light effect in the combined C37:2 and C37:3 values doesn't mean that the unique alkenones are not reacting to light differently, or that the hydrogen isotope ratios of other cellular lipids (and lipid precursors) are not reacting to light.

*Response: When compared to the results from van der Meer et al., 2015 in which the alkenones were measured in the same manner, less fractionation was expected at high light than in the other experiments at much lower light intensities. This was not observed here, in that sense it looks like light intensity does not affect the use of alkenones as a possible paleo salinity indicator. We cannot exclude the possibility the light intensity might affect other compounds, in fact van der Meer at al. 2015 showed that light intensity by itself does affect fractionation in alkenones when E. huxleyi was grown at a constant salinity. This manuscript shows it does not affect the α-salinity relationship.*

In this sense, measuring hydrogen isotopes in all possible lipids is a powerful tool for helping to understand these cellular mechanisms and has tremendous value beyond validating paleoproxies.

*Response: Yes, we agree that measuring other compounds could be useful in helping to understand cellular mechanisms further, and this will be a topic of further study.*

In the end – I am not sure you need to devote an entire section of the discussion to explaining the lack of light effect with only two light levels (and only one lipid measurement) presented. It would be great if this section of the paper could be extended with concentration data or additional lipid isotopes if that is available.

*Response: In fact, we are comparing a few different experiments using E. huxleyi, taking into account different strains, temperatures, experimental set-up, nutrients, growth rates, and light intensities. Given this, we find it surprising that the responses of fractionation to salinity are all statistically similar. Thus, we argue that it is not just two light intensities, but indeed a lot more. As mentioned above, we agree that other compounds could be helpful in elucidating intracellular mechanisms, but our main emphasis with this paper is to discuss the α-salinity relationship with regards to reconstructing paleosalinity based on the specific biomarker lipid C37 alkenones, and how there does not appear to be a difference between the α-salinity relationship for high and low light levels.*

Technical comments: Abstract: First sentence should read "Over the last decade, hydrogen isotope ratios of long chained. . .."

*Response: This has been revised in the manuscript as suggested.*

Line 18. It would be helpful to add "at low light" (or something along those lines) to the end of the sentence "...and independently assess the effects of salinity and alkalinity".

*Revised to: "Batch cultures of the marine haptophyte* E. huxleyi *strain CCMP 1516 were grown to investigate the hydrogen isotope fractionation response to salinity at high light intensity and independently assess the effects of salinity and alkalinity under low light conditions."*

Intro: p.1 line 28: add the word "and" between UK37 and LDI

*Response: This has been revised in the manuscript*

.p.2 line 30: SPM has not yet been defined.

*Response: We have revised this as suggested by anonymous referee #3.*

p.3 line 2: growth phase should include also the Wolhowe et al. (2009) reference.

*Response: This reference has been included.*

p.3 line 7: "created media by evaporation" . . . some culture studies did not use this method: note that salt was added to ultrapure water in Sachs et al. 2017.

*Sachs et al. (2016) do not specify how they changed the salinity. We have revised the manuscript to say, "This factor may be important as the culture experiments (Schouten et al., 2006; M'Boule et al., 2014; Chivall et al., 2014) investigating hydrogen isotopes from alkenones created media of different salinities by evaporation, which changed alkalinity together with salinity in the culture media."*

p. 3 line 13. How would this work? Please explain why the capacity (of environmental water) to take up H+ would, in theory, impact the hydrogen isotope composition of internal cell water and fractionation during synthesis.

*Revised to: "Alkalinity is essentially the ability of water to neutralize acid, which is linked to the amount of $H^+$. $H^+$ is readily exchanged between extracellular and intracellular water, therefore, the amount of $H^+$ could potentially effect the hydrogen isotope composition of intracellular water which is a source of hydrogen for synthesis of organic compounds."*

Methods: How was total alkalinity determined? How was salinity determined? What was the equation to calculate growth rate based on how many cell count readings over how many days? What was the size of the growth vessel for batch cultures and how many liters of culture were maintained inside? Were growth chambers swirled to prevent sedimentation? How many grams of Al2O3 were used and how many ml of solvents for purification? What were the FID and IRMS instrument settings (column, oven temp, reactor temp, gas flow, etc)? Were external standards used to correct/calibrate reported Isodat values referenced to H2 for each run? (you could cite a previous paper with this info if the settings were the same).

*Response: Different alkalinities were created by adding $NaHCO_3$ and $Na_2CO_3$ to increase and concentrated HCl to decrease total alkalinity, which was measured by titration with 0.1 M HCl, and calculated using Gran plots.*
*Salinity was measured using a VWR CO310 Portable Conductivity, Salinity and Temperature Instrument.*
*Growth rate was calculated as the slope of the linear fit of the natural logarithm of cell density (ln[cell density]) in the exponential part of the growth curve.*
*Cells were counted daily over the experimental period of 10-12 days which varied due to differences in growth rates. 600ml of media in triplicate for alkalinity/salinity experiments and 150ml for high light experiments. Smaller volumes were used for the high light experiments because we wanted to ensure that all parts of the culturing vessel would remain under the same constant high irradiance.*
*Yes, growth chambers were swirled to prevent sedimentation.*
*FID and IRMS settings are the same as described in M'Boule et al., 2014.*
*This information has been incorporated in the methods section of the revised version of the manuscript.*

p.5 line 10: "difference" (not "different")

*Response: This has been fixed.*

Results: Several findings that are brought up in the discussion are not mentioned in the results (but should be) including: constant alpha over a range of alkalinities, the weak negative correlation of growth rate and fractionation for both experiments, the lack of light effect. Why not report the uncertainty on the slope and intercept of the least squares linear regressions? If you are using R for your stats, then that is easy to get using R's "summary(lm(x_y))". Although only very slightly biased in this case, it is probably more responsible to report R's Adjusted R-squared as opposed to the Multiple R-squared.
In the first 2 lines of the results the ranges are reported with different dashes ("25 – 35" and then "26-42")

*This has been revised to: "For the alkalinity/salinity experiment, $\alpha_{C37}$ remains relatively constant (0.799±0.003) over the range of alkalinity, but covers a range of 0.776 – 0.824 at constant alkalinity (Table 1) [...] Growth rate is weakly correlated with fractionation for both experiments (Figure 3)." Both are discussed further in the discussion section. Adjusted R-squared values were used.*

p.5 line 20 – "in contrast to previous culture studies" . . . this isn't true – salt was added to fresh water in Sachs et al. 2017.

*Response: This has been revised accordingly; please see previous comment above regarding this.*

Discussion: p.6 line 19 – incorrect reference – Sachs and Kawka 2015 did not find a correlation between growth rate and salinity since growth rate was held constant using continuous cultures – this reference would be more appropriate in the proceeding sentence since they did find a correlation between growth rate and fractionation.

*Revised to: "There is a weak negative correlation of growth rate (µ) with fractionation for both the alkalinity/salinity and high light experiments, $\alpha = -0.0692\mu + 0.8557$ ($R^2=0.25$, $n=24$, $p<0.05$) and $\alpha=-0.1257\mu + 0.8776$ ($R^2=0.35$, $n=14$, $p<0.05$), respectively (Figure 2), which aligns with findings of Sachs and Kawka (2015) who report a negative correlation between growth rate and fractionation, albeit a more strongly correlated relationship."*

p.7 line 6. – missing the word "to" – after the words "could be due"

*Response: This has been fixed.*

p. 7 lines 15-17 this doesn't make much sense – please rephrase – are you trying to say that cell water is the same as extracellular water isotopically and NADPH is more important?

*Revised to: "NADPH is associated with large isotope fractionation values whereas there is less fractionation between extracellular and intracellular water, but both are used as sources of H for synthesis of organic compounds (Schmidt et al., 2003)."*

p.7 line 22 – missing important reference (Luo, Y.-H., Sternberg, L., Suda, S., Kumazawa, S., Mitsui, A., 1991. Extremely low D/H ratios of photoproduced hydrogen by cyanobacteria. Plant and Cell Physiology 32, 897–900) after "Photosynthetic production causes NADPH to be depleted by _600‰ in D when compared to intracellular water"

*Response: Noted, reference has been added to the revised manuscript.*

p.7 line 24 – missing the letter "s" in "OPP pathway cause_"

*Response: Has been changed.*

p.7 line 32 – after the statement "A similar mechanism could be present in E. huxleyi, causing the metabolically reduced NADPH pool to increase relative to other pools, and possibly become a more important source of NADPH for biosynthesis" you might qualify that with something like "if the OPP pathway is present in the same

compartment as alkenone production" (since NADPH doesn't cross organelle membranes). p.8 line 2 – the better reference here is Luo et al. 1991 (provided above)

*Revised to: "A similar mechanism could be present in* E. huxleyi, *causing the metabolically reduced NADPH pool to increase relative to other pools, and possibly become a more important source of NADPH for biosynthesis if the OPP pathway exists in the same location as the site of alkenone synthesis." The reference has also been added.*

Figures: Please specify what the gray polygons are (95% confidence intervals?). Why
no error bars displaying analytical uncertainty on alpha or dD?
Fig.2c is useful for showing the (lack of) relationship between alkalinity and alpha – in this fashion, perhaps an additional figure that displays the (lack of) relationship between light and alpha could also help.
In Fig.2c, does it make sense to include the regression line and confidence intervals here? – might be more useful to use a dashed line or some other visual aid to highlight how 2c is different than 2a and 2b rather than (or in addition to) leaving out the regression statistics.
Fig. 1 might not be necessary, (useful to show that culture water did not have a relationship b/t salinity and dDwater but that is a big figure for a very minor point). Why are a handful of cultures around S=35 15‰ D-enriched relative to the other cultures in the Fig.1a?

*Response: Noted. The D-enrichment is likely due to how the media was created, as this was done separately for the alkalinity/salinity and the high light experiments. We have revised all of the figures to include error bars, the same axis intervals and better captions to more accurately illustrate our data.*

References: p.10 line 12: Is there an extra "."? Check this and other references for common mistakes (superscripts missing in some locations, middle initials incorrect in some locations, missing italics for species names in many locations, too many words in title are capitalized in at least one reference).

*Response: Noted, they have been fixed in the revised manuscript.*

**A. Sessions (Referee)**

This study focuses on the relationship between salinity, alkalinity, and light intensity with H isotope fractionation. The result is to show again that salinity is an important factor, but that alkalinity and light intensity are not. I would characterize this as an important, incremental advance in our understanding of this proxy. The result is not Earth-shaking, but it is an important step forward. The results seem quite clear and unambiguous, and the data analysis and interpretation is convincing. This is a nice study, with little to complain about.

*We would like to thank Dr. Sessions for the comments on our manuscript, which we have taken into consideration and would like to address as "Response:" following the original comment.*

My only general comment is to question why the authors chose to describe culture conditions in terms of alkalinity rather than pH. With [DIC] fixed by equilibrium with atmospheric $PCO_2$, alkalinity and pH are in a sense interchangeable (fixing one uniquely determines the other). Thus the same experiments could be described in terms of either parameter. Alkalinity is probably more popular among oceanographers, but pH is much more widely used among biologists. And I might argue that there is some reason to think that cellular H isotope fractionation depends more on the concentration of H+ (i.e., pH) than on the ability to consume H+ (i.e., alkalinity). So my suggestion is to at least consider describing the first series of experiments as a pH series, rather than an alkalinity series. Or maybe there is a way to gracefully do both.

*Response: We understand that in the natural environment alkalinity and pH are linked, however, in our experiment, we kept pH constant (7.9 ± 0.07) and only changed the alkalinity by adding $NaHCO_3$ and $Na_2CO_3$ to increase and concentrated HCl to decrease total alkalinity in our original media. Revised to: "For batches where alkalinity was changed, pH was kept constant (7.9 ± 0.07)."*

I was curious why a non-calcifying strain of E. hux was chosen. Perhaps it simplifies controlling alkalinity? In any case, it would be worth a few sentences of explanation about why you chose this strain, and how it might relate to strains that are prevalent in the oceans. Is it likely to be representative of strains that produce alkenones in most marine sediments?

*Response: A non-calcifying strain was chosen because we wanted to avoid changes in the total alkalinity of the media caused by the organisms themselves, which has been shown to be the case in the natural environment by Hooligan et al., 1993 (Global Biogeochemical Cycles 7). We agree that a calcifying strain might be more representative of the natural environment, but in order to get a handle on whether alkalinity has an effect on hydrogen isotope fractionation or not, we chose a non-calcifying strain that has been used in previous studies (M'Boule et al., 2014). Furthermore, all previous culture experiments were done with non-calcifying strains, therefore, by using a non-calcifying strain we could better compare our results with those of previous studies. This has been revised to: "A no longer calcifying strain of E. huxleyi, CCMP 1516, was used in these batch cultures. Because the effects of alkalinity on hydrogen isotope fractionation were being assessed, a non-calcifying strain was chosen to avoid changes to the alkalinity of the media caused by the organism; changes that have previously been shown to occur during large blooms of E. huxleyi (Holligan et al., 1993)."*

Section 2.1. Please tell us how you measured (or calculated) alkalinity?

*Response: Alkalinity was measured by titration with 0.1 M HCl, and calculated using Gran plots. (G. Gran: Determination of the equivalence point in potentiometric titrations. Part II, Analyst 1952, 77, 661). This has been revised to: "Alkalinity was determined by titration with 0.1 M HCl and calculated using Gran plots (Gran, 1952; Johansson et al., 1983; Hansson and Jagner, 1973)."*

Page 6, line 25. You say that you performed a statistical comparison, and then that "This showed a strong similarity between slopes..". What does strong similarity mean in statistical terms? They are indistinguishable? Given that the slopes differ between experiments by more than a factor of 2, this is probably more a statement about variability between experiments rather than a constant slope. Seems like the discussion of this 'similarity' could be a bit more nuanced. Differences of a factor of _2 would still make a huge difference in reconstructing seawater salinity, even if they are statistically indistinguishable.

*Response: True. We were referring to the fact that the slopes were not statistically different because we wanted to examine the data as a whole data set, but we realize that this might not actually be useful for reconstructing salinity because of the differences mentioned above. Furthermore, we believe that the individual experiments themselves (i.e., conducted by different people in different labs using different techniques) do play a large role in the observed differences between slopes, as well as different strains of* E. huxleyi *that were utilized. Revised to: "We performed a statistical comparison using ANCOVA between the different $\alpha_{C37}$-salinity relationships for previous* E. huxleyi *cultivation experiments (Schouten et al., 2006; M'Boule et al., 2014; Sachs et al., 2016; Table 2) and our experiments. Sachs et al. (2016) report $\delta D$ values for individual alkenones, thus we used a weighted mean average of the $\delta D_{C37:3}$ and $\delta D_{C37:2}$ values to compare with other results reporting integrated $\delta D_{C37}$ values. The slopes of the $\alpha_{C37}$-salinity relationships are not statistically different from each other (p > 0.05), with the exception of three comparisons: Sachs et al. (2016) was statistically different (p ≤ 0.05) from Schouten et al. (2006), M'Boule et al. (2014) and the Alkalinity/Salinity experiment (Table 2). A possible explanation for the statistical difference between the $\alpha_{C37}$-salinity relationship of Sachs et al. (2016) and the other three experiments could be due to the fact that Sachs et al. (2016) conducted the experiment using chemostats, whereas, the other experiments were batch culture experiments. Growth rate has been shown to effect hydrogen isotope fractionation of alkenones (Schouten et al., 2006; Wolhowe et al., 2009; Sachs and Kawka et al., 2015), therefore could account for the difference between reported fractionation responses to salinity. Although the slopes are statistically similar (p > 0.05), the individual experiments themselves (i.e., conducted by different labs using different techniques) do play a large role in the observed differences between slopes, as well as different strains of* E. huxleyi *that were utilized. Furthermore, the intercepts of the regression models applied to the $\alpha_{C37}$-salinity relationships for the* E. huxleyi *culture data are all significantly different (p ≤ 0.05), i.e. the absolute fractionation differs between the different studies, except for the relationship reported by M'Boule et al. (2014) and our high light experiment. These differences in intercept may be explained by a number of potential factors. One explanation could be due to the different strains of* E. huxleyi *used in the cultivations, as each strain would respond in a similar fashion to salinity changes but fractionate to a different extent. This could be due to differences in fractionation and intracellular sources of hydrogen or differences in lipid synthesis rates".*

Page 7, line 6. The differences in intercepts amount to a range of nearly 78‰. That does not seem (to me, at least) plausible to explain solely by interlaboratory differences. Maybe modify the text to say that "part of" the differences could possibly be attributed to this.

*We have revised this to: "Another explanation for part of the discrepancies in intercepts could be analytical differences between laboratories, i.e. small offsets in measured absolute values of $C_{37}$ alkenones. Inter-laboratory comparison of measured hydrogen isotope values of an alkenone standard could help to eliminate this uncertainty."*

Page 7, lines 20-35. The term "photosynthetically-derived NADPH" struck me as a little odd, especially in contrast to the more precise "pentose-phosphate pathway". Photosynthesis both produces (in photosystem I of the light reactions) and consumes (in CO2 fixation of the Calvin cycle reactions) NADPH. It would thus be more precise to refer to NADPH from the "light reactions of photosynthesis", or to "ferredoxin-NADP+ reductase (FNR) in photosystem 1", etc.

*Revised to: "Photosynthetic production causes NADPH to be depleted by ~600‰ in D when compared to intracellular water (Luo et al., 1991), whereas NADPH produced via the OPP pathway is also depleted compared to intracellular water, but much less than NADPH derived via ferredoxin-NADP+ reductase (FNR) in photosystem 1 (photosynthetically derived) (Schmidt et al., 2003; Maloney et al., 2016)."*

Page 7, line 30-33. I like this explanation, a lot. It is the best one I have heard yet.

*Response: Thanks.*

Page 9, line 4. "At higher light intensities, we expect a larger pool of photosynthetically derived NADPH inside the cell," Do you have direct evidence (either your own, or from a reference) to support this? Photosynthesis is pretty tightly regulated, so my expectation would be that as soon as NADPH levels start to creep up, photons are shunted to nonphotochemical quenching instead of to the photosystems and NADP reduction. In which case, NADPH levels might not depend on light levels. There should be papers about this in the biochemical literature.

*Response: Due to balancing between ATP and NADPH production and consumption within the cell (Walker et al., 2014), NADPH formation dominates at high light levels, whereas ATP synthesis dominates at lower light levels (Beardall et al.), leading to the idea of a larger pool of photosynthetically derived NADPH inside the cell. We have added these references to our discussion of this topic. It now reads: "Due to balancing between ATP and NADPH production and consumption within the cell (Walker et al., 2014), NADPH formation dominates at high light levels, whereas ATP synthesis dominates at lower light levels (Beardall et al., 2003), leading to the idea of a larger pool of photosynthetically derived NADPH inside the cell, which could, in turn, cause differences in hydrogen isotope fractionation during the synthesis of alkenones."*

Page 9, lines 5-10. Larger pool of reduced NADPH could also mean a longer lifetime, and greater D/H exchange.

*Response: Yes, true. We agree.*

Table 1. Can you at least include the initial alkalinity and/or pH for the high-light experiments? It is not essential, just seems weird not to report them given the emphasis on that variable of the rest of the paper.

*Response: Alkalinity and pH were not measured for the high light experiments, since they were performed separately and not originally considered to be discussed along-side the alkalinity/salinity experiment results, but we used filtered North Sea water, so the alkalinity is presumably around 2.3 (Brasse et al., 1999) Please see revised Table 1.*

**Anonymous Referee #3**

The study "Effects of alkalinity and salinity at low and high light intensity on hydrogen isotope fractionation of long-chain alkenones produced by Emiliania huxleyi" by Gabriella M. Weiss, Eva Y. Pfannerstill, Stefan Schouten, Jaap S. Sinninghe DamsteÌA¸, and Marcel T.J. van der Meer is an important step forward in the quest to understand the environmental sensitivities of hydrogen isotope fractionation during lipid biosynthesis in unicellular photoautotrophs. The experimental design, measurements, analysis and interpretations are all high quality, and I have no major criticisms of the work. However, I do have a few suggestions about ways that the manuscript could be improved, which I have outlined below. I recommend publication once the authors have had a chance to consider these, and the input from the other reviewers.
*We would like to thank Anonymous Referee #3 for the constructive comments, which we will take into consideration and would like to address below in italics following the original comment.*

General comments:
I found the discussion on mechanisms of hydrogen isotope fractionation to be well thought out and referenced, but I think that a slightly expanded discussion on the growth conditions that E.hux experiences in the environment, seasonality of lipid production, and effects of growth rate, light, and nutrients, etc. on alkenone production might be helpful to guide the discussion on alkenone H-isotope fractionation. This is a very well-studied organism after all, and we benefit from decades of research on these factors due to the work that has been done for understanding Uk37 and 13C/pCO2 applications. I suspect that these lessons could be applied to the present work a bit more than they are currently.

*Response: This has been revised to include: "Growth rate and irradiance have also been proven to influence total carbon isotope fractionation of alkenones used as a $pCO_2$ proxy (Pagani, 2014 and references therein). Both of these factors are related and seem to play a significant role for isotopic fractionation of alkenones, and the effects remain to be completely understood."*
*As well as, "The fact that the strong $\alpha_{C37}$-salinity response is also identified in* E. huxleyi *grown at high light conditions is important for understanding the influence of light and depth effects (i.e., van der Meer et al., 2015; Wolhowe et al., 2015) on the $\alpha_{C37}$ of alkenones preserved in the sedimentary record. Van der Meer et al. (2015) suggested that at light intensities above 200 μmol photons $m^{-2}$ $s^{-1}$, $\alpha_{C37}$ responds differently to changes in light intensity than below, with a larger reported range in fractionation values at light intensities below 200 μmol photons $m^{-2}$ $s^{-1}$. Wolhowe et al. (2015) also show this trend in $\alpha_{C37}$ measured on alkenones in suspended particulate matter from the Gulf of California and Eastern Tropical North Pacific. Krumhardt et al. (2016) indicate that although haptophyte indicative pigments were high below surface water layers in the subtropical North Atlantic, they were also abundant in the upper 30 m of the water column, especially during spring. Based on these findings and the UK'37 core-top calibration, we can be confident that alkenones preserved in the sediments are largely reflecting surface water temperatures during the time of the year that haptophytes are known to bloom (Müller et al., 1998). Furthermore, haptophytes are thought primarily to bloom at light intensities above 500 μmol photons $m^{-2}$ $s^{-1}$ in the surface ocean (Nanninga and Tyrrell, 1996), leading to the conclusion that the light and depth effects discussed previously (van der Meer et al., 2015; Wolhowe et al., 2015) might not have such a large effect on the $\alpha_{C37}$-salinity response as previously believed."*

In general the paper is careful to specify when discussing alkenones produced by E.Hux from those produced by other haptophytes, but there are a few cases where this isn't clear and I'd recommend clarifying these.

*Response: We have revised this to clarify when discussing* E. huxleyi *or other species each time they are mentioned.*

The quality of the writing is fine for the most part, although I do have a few suggestions and typos that I've outlined below, so I would also just recommend here that it gets read again with this in mind.

Detailed comments:
P.2L1-10 - I would not refer to continental bodies of water as meteoric this way. Meteoric implies precipitation-derived and that dD = 8*d18O +10, and many continental bodies of water are enriched due to evaporation, making them non-meteoric. Also, meteoric lakes and rivers are very fresh, making the statement about "low salinity" a little odd. I might rephrase this sentence to say something like "Therefore, most lakes and rivers that are fed by

precipitation (i.e. meteoric waters) are characterized by a depleted isotopic signature. As these waters drain into the ocean and mix with seawater the result is to lower both the sea surface salinity as well as the water isotope value, as also occurs during direct precipitation on the ocean."

*Response: Thank you for the suggestion, we have revised to: "Therefore, a depleted isotopic signature is found for most precipitation-fed rivers and lakes (i.e. meteoric waters). As these waters drain into the ocean and mix with seawater, the sea surface salinity is lowered, as is the water isotope value."*

P2.L17 - It is a little confusing here to use alpha without really defining it, especially here in this context since the sentence describes hydrogen isotope fractionation decreasing with increasing salinity, meaning an increase in the fractionation factor in this case. I think you don't need to define alpha right here anyway, so I might suggest leaving it until later, at the first actual required use. When the first use and definition do come, I'd also recommend including an equation at that point in line with the normal text because the definition of alpha is always application specific depending on the nature of the isotope system and product/substrate applications in any particular study.

*Response: Noted, in the revised version, alpha has been referred to as hydrogen isotope fractionation until section 2.3, at the end of which the equation for alpha was previously defined.*

P2.L21-27 – Leduc et al., EPSL, 2013 would be another reference that might be worth discussing here in example applications.

*This has been revised to: "Leduc et al. (2013) show a divergence in estimates of sea surface salinity between $\delta D_{C37}$ dervied and planktonic foraminifera derived proxies ($\delta^{18}O_{sw}$ and Ba/Ca ratios) across the Holocene in the Gulf of Guinea, which are attributed to differences in the isotopic ratios of rainfall over the time period."*

General - Throughout the entire manuscript, delta values (of all types) would be better used in the text with the word "value" (e.g. "d18Oforam values from the same region…", as opposed to "d18Oforam from the same region…")

*Response: This has been corrected in the revised version.*

P2. – Acronyms should be defined at first use or not used in my opinion, so on this page "LGM" and "SPM".

*Response: These acronyms have been removed in the revised version and replace with the full names.*

P2-3 – This paragraph might cite Nelson and Sachs, OG, 2014 in the discussion of field studies, and Wolhowe et al., Biogeosciences, 2009, in the discussion of growth phase. Also be sure to be clear about which observations/studies apply specifically to E.Hux and which don't.

*Revised to: "Nelson and Sachs (2014) also tested the use of $\delta D_{C37}$ in North American lakes covering a salinity range of 10-133. In the North American lakes, there is a relationship between $\delta D_{H2O}$ and $\delta D_{C37}$, but no trend between fractionation and salinity (Nelson and Sachs, 2014). These environmental datasets suggest that there are other factors affecting hydrogen isotope fractionation, which complicate the use of $\delta D_{C37}$ as a salinity proxy. Indeed, culture studies have indicated that hydrogen isotope fractionation can be influenced by a number of parameters, i.e., growth rate (Schouten et al., 2006; Wolhowe et al., 2009), growth phase (Chivall et al., 2014), species composition (M'Boule et al., 2014; Chivall et al., 2014), and irradiance (van der Meer et al., 2015)."*

P3.L7 – Reference needed for "the culture experiments"

*Revised to: "This factor may be important as the culture experiments (Schouten et al., 2006; M'Boule et al., 2014; Chivall et al., 2014) investigating hydrogen isotopes from alkenones created media of different salinities by evaporation, which changed alkalinity together with salinity in the culture media."*

P3.L9 – change "on" to "by"

*Response: This has been revised.*

P3.L17 – Reference needed for "most of the culture studies"

*Revised to: "However, some of the culture studies that reported a strong correlation between hydrogen isotope fractionation and salinity were performed at relatively low light intensities (Wolhowe et al., 2009; M'Boule et al., 2014; Chivall et al., 2014)."*

P3.L29 – Not necessarily here, but somewhere in the paper there should be a discussion about why a non-calcifying strain was selected.

*Response: Noted, see comment to Referee A. Sessions. Revised to: "A no longer calcifying strain of E. huxleyi, CCMP 1516, was used in these batch cultures. Because the effects of alkalinity on hydrogen isotope fractionation were being assessed, a non-calcifying strain was chosen to avoid changes to the alkalinity of the media caused by the organisms, which has previously been shown to occur during large blooms of E. huxleyi (Holligan et al., 1993)."*

P4.L1-5 – Somewhere in here it would be good to state the pH of the cultures too.

*Response: Noted, see comment to Referee A. Sessions. Revised to: "For batches where alkalinity was changed, pH was kept constant (7.9 ±0.07)."*

P5.L.19 - change "and therefore" to "and were therefore"

*Response: This has been revised.*

P5.L20 - provide reference when making a comparison to "previous studies"; P5.L21 - remind us here, as well as in the caption for figure 1, why those dDwater values are so high in that one group of samples. Maybe different colors for the modified alkalinity samples in the figure?

*Revised to: "Since the salinity of the media of was not altered by evaporation but by addition of NaCl, in contrast to previous culture studies (Schouten et al., 2006; M'Boule et al., 2014; Chivall et al., 2014), $\delta D_{H2O}$ values were not correlated with salinity due to separate creation of the media (Figure 1)."*

P5.L24 - alpha values should also be discussed in the text using the word "value" so change "a37" to "a37 values".

*Response: This has been revised throughout the manuscript.*

Section 3 - Results sound better when consistently described in the past tense in my opinion (e.g. P5.L24-25 as: "A strong linear relationship between _C37 values and salinity was observed in both experiments). Either way, be consistent about tense use throughout.

*Response: This has been revised throughout the manuscript.*

P6.L10-15 - It would be useful here to provide a reference to what surface ocean light levels are and how these dissipate with depth.

*Revised to: "Ocean surface light levels span a range from zero to over 800 PAR (over 1600 µmol photons m$^{-2}$ s$^{-1}$) (Frouin and Murakami, 2007), and haptophytes not believed to be photoinhibited and primarily bloom at light intensities above 500 µmol photons m$^{-2}$ s$^{-1}$ (Nanninga and Tyrrell, 1996)."*

P6.L15 - "statistically similar" – be quantitative

*Revised to: "The slope of the α-salinity correlation, or fractionation response per unit salinity, is statistically similar (p>0.05) for both the alkalinity/salinity and the high light experiments, based on analysis of covariance (ANCOVA) between the linear regression models fit to each dataset."*

P6.L23-25 - Which individual C37 alkenone from the Sachs16 reference is being used to compare to the mixed C37 alkenone dD values reported in the other studies?

*Response: We used a weighted mean average to compare the separated C37 alkenones measured by Sachs et al., 2016 to the integrated alkenones in our experiments and the previous experiments of E. huxleyi.*
*Revised to: "Sachs et al. (2016) report $\delta D$ values for individual alkenones, thus we used a weighted mean average of the $\delta D_{C37:3}$ and $\delta D_{C37:2}$ values to compare with other results."*

P6.L24 - Chivall14 used a coastal producer, no? The equation is also not listed in table 2. Should this reference be omitted from this list?

*Response: Yes. This has been removed.*

P6.L30 - I think that this issue of production depth/light exposure could benefit from a slightly expanded summary and literature survey. This gets to one of my general comments at the beginning. It might also be useful to comment on where in the ocean one might expect to find light levels that might cause a large H-isotope effect (i.e. <100 umol, based on the van der Meer, GCA, 2015 paper).

*Response: See comment to Anonymous Referee #1. Revised to: "With the exception of the high light experiment, all other culture experiments with E. huxleyi being discussed here were grown between 50-300 $\mu mol$ photons $m^{-2}$ $s^{-1}$ (Schouten et al., 2006; M'Boule et al., 2014; Sachs et al., 2016; alkalinity/salinity experiment). The fact that the strong $\alpha_{C37}$-salinity response is also identified in E. huxleyi at high light conditions is important for understanding the influence of light and depth effects (i.e., van der Meer et al., 2015; Wolhowe et al., 2015) on the $\alpha_{C37}$ of alkenones preserved in the sedimentary record. Van der Meer et al., 2015 suggested that at light intensities above 200 $\mu mol$ photons $m^{-2}$ $s^{-1}$, $\alpha$ responds differently to changes in light intensity than below, with a larger reported range in fractionation values at light intensities below 200 $\mu mol$ photons $m^{-2}$ $s^{-1}$. Wolhowe et al. (2015) also show this trend in $\alpha_{C37}$ measured on alkenones in suspended particulate matter from the Gulf of California and Eastern Tropical North Pacific. Krumhardt et al. (2016) indicate that although haptophyte indicative pigments were high below surface water layers in the subtropical North Atlantic, they were also abundant in the upper 30 m of the water column, especially during spring. Based on the UK'37 core-top calibration, we can be confident that alkenones preserved in the sediments are largely reflecting surface water temperatures during the time of the year that haptophytes are known to bloom (Müller et al., 1998), and haptophytes are thought primarily to bloom at light intensities above 500 $\mu mol$ photons $m^{-2}$ $s^{-1}$ in the surface ocean (Nanninga and Tyrrell, 1996), leading to the conclusion that the light and depth effects discussed previously (van der Meer et al., 2015; Wolhowe et al., 2015) might not have such a large effect on the $\alpha_{C37}$-salinity response as previously believed."*

Section 4.2 - Describe the statistical similarities in slopes and differences in intercepts in quantitative terms. What thresholds were applied?

*Revised to: "We performed a statistical comparison using ANCOVA between the different $\alpha_{C37}$-salinity relationships for previous E. huxleyi cultivation experiments (Schouten et al., 2006; M'Boule et al., 2014; Sachs et al., 2016; Table 2) and our experiments. Sachs et al. (2016) report $\delta D$ values for individual alkenones, thus we used a weighted mean average of the $\delta D_{C37:3}$ and $\delta D_{C37:2}$ values to compare with other results reporting integrated $\delta D_{C37}$ values. The slopes of the $\alpha_{C37}$-salinity relationships are not statistically different from each other ($p > 0.05$), with the exception of three comparisons: Sachs et al. (2016) was statistically different ($p \leq 0.05$) from Schouten et al. (2006), M'Boule et al. (2014) and the Alkalinity/Salinity experiment (Table 2). A possible explanation for the statistical difference between the $\alpha_{C37}$-salinity relationship of Sachs et al. (2016) and the other three experiments could be due to the fact that Sachs et al. (2016) conducted the experiment using chemostats, whereas, the other experiments were batch culture experiments. Growth rate has been shown to effect hydrogen isotope fractionation of alkenones (Schouten et al., 2006; Wolhowe et al., 2009; Sachs and Kawka et al., 2015), therefore could account for the difference between reported fractionation responses to salinity.*

*The intercepts of the regression models applied to the $\alpha_{C37}$-salinity relationships for the E. huxleyi culture data are all significantly different ($p \leq 0.05$), i.e. the absolute fractionation differs between the different studies, except for the relationship reported by M'Boule et al. (2014) and our high light experiment. These differences in intercept may be explained by a number of potential factors. One explanation could be due to the different strains of E. huxleyi used in*

*the cultivations, as each strain would respond in a similar fashion to salinity changes but fractionate to a different extent. This could be due to differences in fractionation and intracellular sources of hydrogen or differences in lipid synthesis rates. Another explanation for part of the discrepancies in intercepts could be analytical differences between laboratories, i.e. small offsets in measured absolute values of $C_{37}$ alkenones. Inter-laboratory comparison of measured hydrogen isotope values of an alkenone standard could help to eliminate this uncertainty."*

P7.L6 - change "due differences" to "due to differences"

*Response: This has been fixed.*

P7.L6 - As written, "different sources of hydrogen" is probably not the best language. I gather that the implication invokes biochemical mechanisms relating to the routing of hydrogen during biosynthesis, but the way it is currently makes it sound like they are accessing different source water, which is probably not the intention.

*Revised to: "This could be due to differences in fractionation and intracellular sources of hydrogen or differences in lipid synthesis rates."*

P7.L5-9 - what about chemostats vs. batch cultures? That probably deserves a mention somewhere in here in comparing to Sachs16.

*Revised to: "A possible explanation for the statistical difference between the $\alpha_{C37}$-salinity relationship of Sachs et al. (2016) and the other three experiments could be due to the fact that the Sachs et al. (2016) experiment was conducted using chemostats, whereas, the other experiments were batch culture experiments."*

P7.L13-15 - Nelson and Sachs, GCA, 2014 would be worth including in this list of references

*Response: Reference has been added. See next comment.*

P7.L14 – I would specify "algal" or "unicellular" photoautotrophs, or include references to alpha-salinity relationships in plants (Aichner et al., OG, 2017; Ladd and Sachs, OG, 2012).

*Response: Revised to: "The effect of salinity on hydrogen isotope fractionation seems to be a general feature recorded in alkenones, fatty acids, sterols, phytene and diploptene produced by algal photoautotrophs (Heinzelmann et al., 2015; Schouten et al., 2006; Sachse and Sachs, 2008; Sachs and Schwab, 2011; Nelson and Sachs, 2014)."*

P8.L22 - change "by OPP" to "by the OPP"

*Response: This has been revised.*

Figure 1 - I suggest labeling the panels directly on the graphs to allow them to be read without looking at the caption. . .. ... I also would personally prefer if the graphs were the same width, and were aligned with each other. I'd also use the same x-axis scale for both, and would note the y-axis scale differences in the caption. . .. State clearly that the gray shaded areas are confidence intervals (they are, right?) and provide the
threshold that was used to define these in the caption
Figure 2 - The font in the axis labels, as well as the plotted symbols look like they were compressed vertically. Can this be fixed so they don't look squished like this? . .. .. ... label the experimental design directly on panels a and b, or consider adding this information using a legend to indicate symbol color. . .. . . see my comment about confidence intervals as related to figure 1 above.
Figure 3 - See my comment about confidence intervals as related to figure 1 above.

*Response: Please see revised figures. Labels have been added to graphs themselves and axis scales have been fixed. Error bars have been added. Captions have been fixed to give more detailed and accurate information about the parameters illustrated in the graphs.*

Anonymous Referee #4

The paper by Weiss et al. presents new data from a laboratory experiment aiming to clarify whether and how strong salinity and light intensity affect the hydrogen isotope fractionation during alkenone biosynthesis. Such results pave the way towards an application of algal lipid biomarker hydrogen isotope ratios as a paleosalinity proxy. While similar experiments have been conducted before and salinity and light intensity have been found to affect the hydrogen isotope fractionation, results from the current study test in particular the effect of alkalinity (which can change independently of salinity) on the isotope fractionation. It therefore adds to the understanding of how representative the previous findings from laboratory cultures are for the natural environment. The study finds that alkalinity does not affect the isotope fractionation and finds similar relationships between isotope fractionation and salinity as observed in previous studies. They also find that changes in light intensity do not change the relationship between salinity and isotope fractionation. These results provide a more robust base to use alkenone D/H ratios as a paleosalinity proxy and may therefore help to identify the actual cellular mechanism responsible for the observed changes in fractionation. While not representing groundbreaking new insights, the study adds to the growing body of literature on this subject. The study is well designed and interpretations are supported by the data. I believe this study should be published after some minor changes. In particular I suggest some clarification of statistical data treatment and a few more detailed descriptions of the experimental setup.

*We would like to thank anonymous referee #4 for their constructive comments and will address them as "Response:" following the original comment.*

General comments: In the study a non calcifying strain of e.hux was used. The authors discuss this to some degree, but a bit more detailed discussion, on how representative these results would be for the natural marine environment, where mostly calcifying strains produce the alkenones, should be part of the discussion.

*This has been revised to: "Because the effects of alkalinity on hydrogen isotope fractionation were being assessed, a non-calcifying strain was chosen to avoid changes to the alkalinity of the media caused by the organisms, changes that have previously been shown to occur during large blooms of E. huxleyi (Holligan et al., 1993)."*

It appears that the statistical data treatment was done using the three replicate data points as individual datapoints – I think it would make more sense to calculate the mean of the replicates and present the standard error of the mean for each treatment. This applies to the actual slope and intercept calculations as well as for the figures and the estimation of the error of the actual regressions (i.e. the shaded area around the regression lines in the figures), see also below.

*Response: Yes, we could combine the data points for the statistical analyses, but we kept them as individual data points because they were separate culture flasks (3 flasks for each variable) with slightly different growth rates. We did average the duplicate isotope measurements.*

The figures could need some more explanation, in the text but also the figure captions. See detailed comments below.

Detailed comments: P6 line 30-31: Can you separate this sentence into 2? It conveys important information, but sounds a bit awkward.

*Revised to: "Since summer blooms occurring under high light conditions in the surface ocean are the most probable source of alkenones preserved in the sedimentary record, the light and depth effects discussed previously (van der Meer et al., 2015; Wolhowe et al., 2015) might not have such a large effect on the $\alpha_{C37}$-salinity response as previously believed."*

P7 line 3-4: Can you mention by how much the intercepts from the other studies vary? I believe it would be instructive to present the data from the current study and previous data from the literature in one graph, see comment below (Table 2).

*Response: Yes, please see supplementary Figure 1.*

P7 line 9: Header for this section does only mention salinity but the second half of the paragraph deals with light intensity. Either separate the paragraph into 2 or mention light intensity in the headline.

*Revised to "4.3 Potential mechanisms for salinity and light responses"*

P7 line 14: In the cited studies not only alkenones, fatty acids and sterols were analyzed, also alkanes and isoprenoids if I remember correctly. I think it would be important to mention that in all these compound classes similar salinity effects have been observed. This is important to identify the underlying mechanism.

*Response: Yes, Sachse and Sachs (2008) also measured phytene and diploptene. We have revised this to: "The effect of salinity on hydrogen isotope fractionation seems to be a general feature recorded in alkenones, fatty acids, sterols, phytene and diploptene produced by algal photoautotrophs (Heinzelmann et al., 2015; Schouten et al., 2006; Sachse and Sachs, 2008; Sachs and Schwab, 2011; Nelson and Sachs, 2014)."*

P8 line 20-21: Interesting hypothesis. Would this hold some advantage for the cell, i.e. using more OPP derived NADPH under higher salinity? Or could this be the result of less water exchange (extracellular with intracellular)?

*Response: Danevčič and Stopar (2011) found a more active pentose phosphate cycle at high salinity in Vibrio sp., and a 10-fold increase in production of L-proline, an osmoregulating amino acid. Up regulating the OPP derived NADPH would be tied to this increase in L-proline, "the production of L-proline, therefore, increases the ratio of intracellular NADP/NADPH, which regulates carbon flux through the oxidative pentose phosphate pathway." (Danevčič and Stopar, 2011). L-proline increases to help continue growth and biosynthesis at higher salinities in* Vibrio sp. *We think something similar could also be happening in* E. huxleyi. *Revised to: "Another explanation for the observed significant correlation with salinity at both high and low light intensity could be that the cell synthesizes alkenones in a closed cell compartment, similar to the 'coccolith vesicle-reticular body' in which coccoliths are formed (Wilbur et al., 1963; Sviben et al., 2016), where the amount of NADPH used for biosynthesis is regulated and the fraction NADPH derived from the OPP pathway into the closed compartment increases with increasing salinity. Danevčič and Stopar (2011) found a more active pentose phosphate cycle at high salinity in* Vibrio sp., *and also that intracellular production of L-proline, an osmoregulating amino acid, increased. The advantage of up regulating the OPP derived NADPH would be tied to this increase in L-proline, which helps*

*continue growth and biosynthesis at higher salinities in* Vibrio sp. *It is possible that something similar to this could also be happening in* E. huxleyi.*"*

Page 8 in general: This is a good summary of the hypotheses being discussed for the observed salinity-fractionation relationship. Except a few points (see above) these have all been proposed in previous papers which have identified the salinity-fractionation dependency. This could be mentioned more explicitly. I suggest to give credit to these papers here, for example in the section about osmolytes the first papers proposing this idea as a factor for the observed change in fractionation, should be cited.

*Response: We have added the references for the discussions already proposed in previous papers. It now reads*: *"In addition to higher abundance of NADPH generated by the OPP pathway at higher salinities, cells also could produce more D-depleted compounds, osmolytes for instance (Dickson et al., 1982; Sachs et al., 2016 and references therein; Sachse et al., 2008), which would leave the intracellular NADPH pool more enriched, which would result in D enrichment of other biosynthetic products such as alkenones."*

Figure 1a: Can you briefly explain, why the culture media water dD values at salinity of 35 are so different from the rest?

*Response: The difference has to do with the way the media were made, as the media were created separately for the alkalinity/salinity and high light experiments. In the revised manuscript it now reads: "Since the salinity of the media of was not altered by evaporation but by addition of NaCl, in contrast to previous culture studies (Schouten et al., 2006; M'Boule et al., 2014; Chivall et al., 2014), $\delta D_{H2O}$ values were not correlated with salinity (Figure 1). Furthermore, the media was created separately for each experiment, causing differences between the original $\delta D_{H2O}$ values seen in Figure 1."*

Figure 1b: I suggest to use the same scale on the x and y axis as in a) Figure 2: also here I suggest to use the same scaling of the x and y axis (at least for salinity). I think that statistically it would make more sense to use the mean of the replicates and their standard deviation for the plots and also to estimate the error of the regression line (standard error of the mean).

*Response: We have fixed the figures but prefer to keep the statistical analyses on the individual points as mentioned above.*

Figure 2c: Can you briefly explain the alpha variability at an alkalinity of 2.5?

*Response: Yes, this is the salinity effect observed in previous culture studies with* E. huxleyi, *seen in our experiment when alkalinity was constant but salinity was varied.*

Figure 3: also here, I suggest to sue the same axis scaling (both for alpha and growth rate and salinity). Clearly, and this is the main point of the paper, salinity has a much stronger effect on isotope fractionation compared to growth rate and this would be easily visible in the graphs, when the same axis scaling is used. Also, if a regression line is plotted through the data, you imply a statistically significant correlation. Is that so in all cases, and if so, then you should present the statistical parameters (p value). If it is not statistically significant, no line should be plotted through the data.

*Response: Noted, we have fixed these figures.*

Table 2: I think it would be useful to see these data compared to the data from the current study in a graph.

*Response: We have plotted the data to give a more visual representation of the comparisons as supplementary Figure 1.*

[revised manuscript text omitted]